



# Low cost, multiscale and multi-sensor application for flooded areas mapping.

Daniele Giordan[1], Davide Notti[1], Alfredo Villa[2], Francesco Zucca[3], Fabiana Calò[4], Antonio Pepe[4], Furio Dutto[5], Paolo Pari[6], Marco Baldo[1], Paolo Allasia[1]

[1]National Research Council of Italy, Research Institute for Geo-Hydrological Protection (CNR-IRPI), Strada delle Cacce 73, Torino 10135. Italy;

[2]ALTEC S.p.A., Torino, 10146, Italy;

[3]Department of Earth and Environmental Science, University of Pavia, Via Ferrata 1, 27100 Pavia, Italy

[4] National Research Council of Italy, Institute for the Electromagnetic Sensing of the Environment (CNR-IREA), Via Diocleziano 328, Napoli 80124, Italy;

[5] Civil protection Service of Torino, Grugliasco, 10095, Italy;

[6]Digisky S.r.l., Torino, 10146, Italy.

*Correspondence to*: Davide Notti (davide.notti@irpi.cnr.it)

## Abstract

Flooded areas mapping and estimation of maximum water height are important elements for a first damages evaluation, civil protection interventions planning and detection of areas where remedial are more needed.

In this work, a methodology for mapping and quantifying flood severity over plain areas is presented and discussed. The proposed methodology considers a multiscale and multi-sensor approach using free or low cost data/sensors. We applied this method to November 2016 Piemonte (NW Italy) flood. We first mapped flooded areas at basin scale using free satellite data from low to medium-high resolution using both SAR (Sentinel-1, Cosmo-Skymed) and multispectral sensors (MODIS, Sentinel-2). Using very- and ultra- high resolution images from the low-cost aerial platform and Remotely Piloted Aerial System, we refined the flooded area and we detected the most damaged. The presented method considers both urbanized and not urbanized areas. Nadiral images have several limitations in particular in urbanized areas, where the use of terrestrial images solved this limitation. Very- and ultra-high resolution images have been processed with Structure from Motion (SfM) for the realization of 3-D models. These data, combined with available digital elevation model, allowed us to obtain maps of flooded area, maximum water high and damaged infrastructures.

## 1 Introduction

Floods are among the natural disasters that cause major damages and causalities (Barredo, 2007).

Mapping and modelling areas affected by floods is a crucial task in order to: i) identify the most critical areas for civil protection actions ii) evaluate damages, iii) and make a correct urban planning. (Amadio et al., 2016). In order to make a precise quantification of damages, a detailed mapping of flooded areas is required, with a good estimation of water level and flow



velocity (Arrighi et al., 2013; Luino et al, 2009; Merz, et al., 2010; Kreibich, 2009). Advances in remote sensing and geotechnology have introduced the possibility, in last years, of having rapid maps and models during or little time after a flood event (Copernicus Emergency Management Service (ⓒ European Union, 2012-2017)). With satellite remote sensing data, it

is possible to map flood effects over wide areas at different spatial and temporal resolution using multispectral (Brakenridge, et al., 2006; Gianinetto et al.,2006; Nigro et al., 2014; Wang et al; 2012; Yan et al., 2015; Rahman and Di, 2017) or Synthetic Aperture Radar (SAR) images (Boni et al 2016; Mason et al 2014; Guy et al 2015; Refice et al 2014; Pulvirenti et al; 2011; Clement et al, 2017; Brivio et al; 2002). A good description of main methodologies used to map flood with satellite data has been published by Fayne et al (2017). In addition, the increasing availability of free-of-charge satellite data with global

coverage (e.g. Sentinel-1 and -2 from ESA, Landsat and MODIS satellites from NASA) makes possible analyses of flooded areas with low cost solutions. Flood mapping and damages assessment is also an important issue for European Communities authorirhyes that support projects like the Copernicus Emergency Management Service mapping (EMSR) and the European Flood Awareness System (EFAS), which manage the activation procedure to acquire satellite data over the areas affected by a natural hazard. Further detail about different experiences in flood mapping in Europe are described in De Moel et al, (2009)

and Paprotny at al, (2017).

In urban areas, remote sensing data are often less efficient in the detection of flooded areas, especially if images acquired during the maximum of inundation are not available. A partial solution could be the use of a Remotely Piloted Aerial System (RPAS) (Perks et al., 2016; Feng et al 2015), that are usually able to acquire ultra-high resolution images over small areas. The quantification of the maximum water level caused by the inundation is an important parameter in particular in urban areas

because it can supply the damages estimations and support civil protection operations (Luino et al., 2009; Bignami et al., 2017). Very often, nadiral remote sensed platforms cannot be able to the definition of the level of water and, for this reason, field surveys and ground-based photos are still necessary. A possible solution is the use of models for the estimation of water depth based on DTM or hydraulic model (Bates and De Roo, 2000; Segura-Beltrán et al, 2016), but ground truth validation is needed. Recent developments of computer vision applications like Structure from Motion (SfM) (Snavely, 2008) made this

system a possible valid alternative for the acquisition of a 3D dataset that can be useful for the definition of the level of water of flooded areas using terrestrial or aerial image acquisition systems. 3-D models derived from SfM are nowadays used for geomorphological applications (Westoby et al., 2012) and for flood mapping. This second application is often assisted using precise DTM derived from Lidar (Smith et al; 2014; Meesuk et al; 2015, Costabile et al., 2015). Particular applications of SfM can be used to make 3-D models of façades and acquire a dataset useful for the identification of marks left by water. The use

of combined low cost systems able to acquire nadiral images and terrestrial oblique pictures is important for the acquisition of



a dataset that can be used for the definition of water level and damages of flooded areas especially in an urban environment (Griesbaum et al., 2017).

Finally, geolocated photos or information deriving from internet and social media (Rosser et al., 2017; Fohringer et al., 2015) or by a volunteer geographic information (Hung et al., 2016, Schnebele and Cervone, 2013) can be very useful for improving
the mapping of flooded areas.

In this work, we present a smart multi-scale and multi-platform methodology developed for the identification and mapping of flooded areas. The methodology has been tested in two areas struck by the flood occurred in Piemonte (NW Italy) in November 2016. The paper presents different case studies that are representative of urban and or not urbanized areas.

## 2 Study areas

Piemonte region is located in NW Italy and most of the territory is inside Po river drainage basin. (Fig 1 A). The Alps range surrounds the region from North to South-West with an elevation higher than 4000 m asl. In the southern sector, Ligurian Alps and Apennines range present lover elevation (1000-2000 m asl) and separate Piemonte from the Liguria sea. At East is open to Po river plain. This orographic setting tends to amplify effects of some particular meteorological conditions like strong and slow moving cyclones located at west of Italy that cause a wet flow from South / East that is blocked by Alps range. This
meteorological configuration causes heavy rainfalls especially in autumn when the warm Ligurian sea is a source of additional energy and humidity (Buzzi et al.,1998; Pinto et al., 2013). In the last 30 years, 4 strong major floods hit this region: September 1993 (Regione Piemonte, 1996,), November 1994 (Luino, 1999),  October 2000 (Cassardo et al., 2013) and November 2016 (ARPA Piemonte, 2016)

In November 2016, a severe flood hit the Piemonte region (NW of Italy). In several areas of Piemonte, in the period 21 – 25
November 2016 different rain gauges registered an amount of rainfall up to 600 mm that represents the 50 % of the main annual precipitation (Fig. 1 B). The basin of Po and Tanaro rivers were the most affected by the flood that was very similar, in terms of rainfall distribution and river discharge, to 1994 event, which is considered one of the most destructive occurred in last decades (Luino, 1999). This time, the event caused huge damages, activated numbers of landslides and debris flows and caused the inundation of large areas. The civil protection system managed the emergency and the number of victims was
strongly reduced with respect to 1994 event that caused 70 victims. The 2016 flood caused a victim in Chisone valley, not far from Torino.

The presented case study area is located in the Po plain south of Turin city (Fig. 1 C). This area is mainly occupied by intensive agricultural activity and urban areas especially located in the northern part. In the southern, close to Torino, many industrial



and commercial areas were built in last decades nearby rivers. From the geomorphological point of view, the actual plain (Fig.

1 B and Fig 1 C) correspond to the fill of Plio-Pleistocenic Savigliano basin (S.B.), delimited by western Alps, Turin C) Hills (T.H.) and Poirino Plateau (P.P). To west is possible to find alluvial fans of Chisone, Pellice and Chisola streams (Carraro et al., 1995). The plain is marked by the terraces that delimit of actual Po valley with evident relict geomorphology like paleo-meander. The anthropic influence is remarkable with like quarry lake, revetments and embankment that constrain riverbeds (Fig. 1C). The geomorphology is a key factor that derived flooded area shape and the water height.

This area was affected by the flooded of Po River and other tributaries, in particular, Chisola and Oitana streams causing several damages. The Po river between Carignano and Turin stations reached a maximum discharge of 2000-2200 m$^3$/s in the late evening of 25 November 2016. The main water discharge of this monitoring station in November is 70 m$^3$/s. The Chisola stream registered a discharge of 200 m$^3$/s (November average 17 m$^3$/s) near Moncalieri in the afternoon of 25 November (ARPA Piemonte, 2016).

Inside this area (Fig. 1 C) we focused our attention in particular on two sites were high resolution data were acquired:

•        The village of Pancalieri, located in on the left side of Po river, just after the confluence with Pellice river. In this area it is evident the presence of ancient meanders of Po river that were reactivated by the flood with damages to some settlements and destruction of communications roads.

•        The town Moncalieri (about 60'000 inhabitants) is located south of Turin in a very manmade environment. This area

was flooded by Chisola stream on the late morning of 25 November partly due to the collapse of some sections of river embankment. The water interested many residential, service and industrial areas with a maximum water heigh of 1.5 – 2 m. Part of Moncalieri municipality was also flooded by Po River in the evening of 25 November, with other damages to commercial and industrial infrastructures.

The activation of Copernicus Emergency Management Service (ⓒ 2016 European Union) EMSR-192

(http://emergency.copernicus.eu/mapping/list-of-components/EMSR192) allowed to map flooded areas (delineation maps) using Radarsat-2 and Pleiades images in the most critical areas of Piemonte and in some areas like Moncalieri also map of damages (grading maps) were produced. However, the delineations maps available represent the automatic flooded area at the moment of image acquisition, and generally not at the maximum extension. In the case of Piemonte, they cannot be used for exhaustive modelling and damages evaluations. The preliminary estimation of damages to buildings made by the municipality

of Moncalieri is about 50 million of € (M€) for industrial buildings and 13 M€ for residential buildings and others 6 M€ for damages to other goods (http://www.comune.moncalieri.to.it/flex/cm/pages/ServeBLOB.php/L/IT/IDPagina/3669).



## 3 Materials and Methods

The aim of this study is the definition of a possible methodology for the identification and mapping of flooded areas using low
cost solutions. For this reason, we have combined and compared data from different sensors, and we used different approaches
for flood mapping some already tested in literature from long time and others more innovative and experimental. We first
introduce the concept of 'pre-flood', 'co-flood' and 'pre/post-flood' data. Co-flood data are collected around the time of
maximum inundation while pre/post-flood data are acquired before or after the flood maximum. In the first case, the mapping
of flooded areas is obviously easier, but the acquisition of co-flood images could not be always possible.

Our study considers both urbanized and not urbanized areas. Using a multi-scale approach, we developed a methodology (Fig.
2) that considers the progressive use satellites and then high and ultra-high resolution systems for the acquisition of a dataset
that can be used to support the identification of water level reached by the flood and occurred damages.

The first identification of the flooded area can be done using satellites results and in situ information coming from the civil
protection system that collects reports from local authorities (co-flood phase). This first identification phase is mandatory to
have a quikly indication of the involved area and to plan more detailed acquisitions.

The second phase is aimed to acquire a high definition dataset that can be used for a detailed mapping of the flooded area. For
this step is required a system able to fly on demand over large areas and acquire a RGB /multispectral dataset with a resolution
of 10-20 cm/pixel. The high-resolution map obtained during this phase can be used for the identification and map of flooded
areas with a good detail. The resolution of the ortophoto can also support the identification of critical elements like damaged
infrastructures: bridges, levees, streets and urban areas involved in the flood. The map of most damaged sectors can be obtained
merging civil defence reports and the analysis of acquired ortophotos. The identification of most critical sectors is important
for a preliminary evaluation of occurred damages and for the emergency planning of first remedial actions.

On the most critical sectors, especially in urban areas, it is possible to acquire ultra-high resolution dataset (2-5 cm/pixel) that
can be used for the quantification of damages or for detailed mapping of flood markers. This third phase can be done using
Remotely Piloted Aerial Systems (RPAS) or terrestrial systems. This last phase is aimed to quantify the flood severity. In our
test, we started from the use of nadiral images acquired by airplanes and RPAS, but we immediately realized that in urban
areas this approach can be not sufficient for the mapping of flooded limits and the identification of damages. One of the most
important data is the water level reached by the flood that is often visible only on façades of buildings. The identification and
mapping of water level markers on façades are mandatory for a correct reconstruction of what happened. To obtain a 3D
representation of urbanized flooded areas, we decided to integrate terrestrial and aerial images using SfM algorithm.

In the following chapters, we present the acquired datasets of different phases (Table 1). All the proposed systems are low cost
solutions that could be adopted by national/regional Authorities with limited efforts.

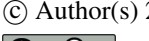



### 3.1 Flood mapping at regional scale with satellite data

The developed methodology is based on a multi-scale approach that starts from the use of low-resolution regional scale satellite images. The use of different available satellites images can support the identification of flood effects at low resolution over large areas and at higher resolution at local scale. The choice of the most appropriate satellite data depends on different factors: i) characteristics of the study area, ii) spatial resolution, iii) revisit time, iv) time of acquisitions respect to the moment of maximum inundation, v) availability and cost of images.

In this paper, we considered only free of charge images to assure a low cost approach. For every considered dataset, we produced a map of the flooded area that represents the synthesis of remote sensing data and geomorphological evidence from 5-m DTM available from Regione Piemonte. We use a visual-operator approach to map flooded areas as resulted more precise than automatic classifications especially in the case of post-flood images. In table 2 are reported the satellite data in relation to the flood phase. We considered as co-flood data all the images acquired between early November 25 (start of the first inundation) and the evening of November 26, 2016 (withdrawn of water).


### 3.1.1 SAR data

SAR instruments work in all-weather conditions and in night time, thus ensuring a high observation frequency and increasing the opportunity to provide data in correspondence at the flood event.

**l) Co-Flood mapping.** If data are available during the maximum flooding phase, it is possible to accurately map the affected
area using high resolution SAR images as those acquired by the TerraSAR-X (Giustarini et al., 2013) and COSMO-SkyMed (Refice et al., 2014) satellites. In particular, the identification of the flooded area is performed by analysing the SAR backscattering, which generally shows low values in water-covered areas. In our analysis, we used a SAR image acquired by the X-band COSMO-SkyMed satellites constellation (wavelength ~ 3 cm) on 25 November (05:05 UTC acquisition time). Data has been provided free-of-charge by E-Geos and Italian Space Agency (ASI) in a quick-look preview format with a 60
m x 60 m resolution. The EMSR service used also Radarsat-2 images for flood mapping. The available COSMO-SkyMed image has been classified into three main land cover classes: water-covered areas, i.e., flooded (low amplitude), urban areas (high amplitude) and soil/vegetation (intermediate amplitude). The analysis with such data points out the relevance of co-flood images for a fast mapping of flooded areas. We remark that it is not possible to know *a-priori* if a co-flood image will be available during the maximum of the flood event, however the short revisit times achieved by the new generation of SAR
satellites can significantly increase the possibility to collect co-flood data.





**II) Post-flood mapping**. We also performed a post-flood mapping by exploiting data acquired by the Sentinel-1 mission which is composed of a constellation of two satellites, Sentinel-1A and Sentinel-1B, launched on 2014 April 3, and 2016 April 25, respectively. Sentinel-1A satellites have been designed to acquire C-band SAR data in continuity with the first-generation ERS-1/ERS-2 and ENVISAT mission, developed within the European environmental monitoring program Copernicus. The Sentinel-1A SAR operates at 5.405 GHz and supports four imaging modes providing images with different resolution and coverage (Torres et al., 2012). We used the Interferometric Wide Swath Mode (IW) acquisition mode by employing the Terrain Observation by Progressive Scans (TOPS). The IW TOPS mode is the main mode of operations for the systematic monitoring of surface deformation and land changes (De Zan and Monti-Guarnieri, 2006). This acquisition mode provides large swath widths of 250 km with a spatial resolution of 5 m × 20 m (IW). The repeat cycle of the twin Sentinel-1A/B constellation is reduced to 6 days.

For our analysis, we have acquired two IW Sentinel-1A/B images collected over the study area, in VH polarization along the satellite descending passes. In particular, we have exploited data acquired after (on November 28, 2016) and before (November 22, 2016) the flooding event (see Table 3).

SAR data, provided in the Single Look Complex (SLC) format, have been first radiometrically calibrated in order to convert the digital number (DN) values into corresponding backscattering coefficients, i.e., sigma naught ($\sigma^o$) values, which contain information on the electromagnetic characteristics of the surface under investigation. Subsequently, calibrated SAR data have been multi-looked with one look in the azimuth direction and four looks in the range one, and finally geocoded, by converting the maps from radar geometry into Universal Transverse Mercator (UTM) coordinates (zone 32T).

After these pre-processing steps, in order to detect land surface changes induced by flooding, we have computed the difference between the post- and the pre- flooding geocoded backscattering coefficient images, and produced the map of the temporal variation of the surface backscattering ($\Delta\sigma^o_{post\text{-}pre\text{-}flooding}$).

### 3.1.2 Multispectral satellite data.

In this category, we considered both low and medium resolution images. Unfortunately, we found co-flood images only for low-resolution images.

**I) Medium-Low resolution satellite data**. MODIS (Moderate Resolution Imaging Spectroradiometer) is a system of two sun-synchronous, near-polar orbiting satellites called Aqua and Terra that daily acquire images all over the World (Justice et al., 1998). Terra acquires images in the late morning while Aqua in the early afternoon, satellites have also a night time pass when they acquire in the thermal bands. This repeat frequency does not occur along the same ground track. The ground track repeat



cycle is every 16 days. This allows detecting with more probability flood over wide areas when they still flooded and not covered by cloud. We searched the available images from Earthdata portal of NASA (https://worldview.earthdata.nasa.gov) and we found an image of Aqua satellite acquired the 26 November 2016 (The Terra image is too cloudy). We used the 6-bands products with a spatial resolution from 250 to 500 m that range from visible to near infrared (NIR) and shortwave infrared (SWIR) (Table 4). For the elaboration, we used a spatial resolution of 500 m/pixel. To have a benchmark of the non-

flooded situation, we considered also the Aqua satellite image of 12 November 2016, which we compared with the image taken during the flood. For the identification of flooded areas, we make the following elaborations:

  a)  False colour image made with combinations of 7-2-1 bands for a visual interpretation of flooded areas;

  b)  Variation of Modified Normalized Difference Water Index MNDWI$_{var}$ (Equation 1). The MNDWI allow detecting water masses or soil moisture. In literature, different combinations for this index are presented and discussed (Xu,

2006; Zhang et al., 2016; Gao, 1996). In our study, we used the ratio between B1 (red band) and B7 (Short Wavelength Infrared - SWIR). The difference with a non-flooded situation can be used for identifying changes in soil moisture.

$$MNDWIvar = MNDWIpost - MNDWIpre \qquad \text{where } MNDWI = \frac{(B1-B7)}{(B1+B7)} \ (1)$$

  c)  Supervised maximum likelihood classification of co-flood image made with SAGA GIS. We manually defined the training areas with main land use typology visible on the image, and then we extracted the category area covered by

water or wet land that correspond to the flooded area. This type of methodology has already been used in literature to map flooded areas as described in Ireland et al., (2015)

**II) Medium-high resolution satellite data**. Medium-high resolution multispectral satellites (e.g. Sentinel-2 a and b or Landsat 8) have a longer revisit time (from 5 days for Sentinel-2 constellation to16 days for Landsat-8) and it is more difficult to have

images at the same time of the maximum flood and cloud free. However, by comparing two images before and after a flood event it is possible to calculate the variation of different indexes related to change in reflectance of the soil or/and of the vegetation. In this way, it is sometimes possible to map the flooded area indirectly (post-flood mapping).

In our study area, we used images taken by Sentinel-2 before the flood (2016, November 11) and after (December 1). Sentinel-2 has some bands at 10-m of spatial resolution and some bands and 20-m of spatial resolution resumed in Table 5. To detect

flooded area, we first made a visual interpretation using images with different bands composition of post-flood data. The comparison of considered images allowed to calculate the difference between two indexes:

1. The variation of NDVI (Normalized Difference Vegetation Index - equation 2). The NDVI calculated with 10 m of spatial resolution images Sentinel-2 using the near infrared band (NIR - B8), and the red band (B4). The NDVI is related to the activity



of vegetation and it is possible to identify the decrease of NDVI values as an effect of inundation on vegetation (Ahamed et
al., 2017). Using this approach, it is possible to indirectly map flooded areas.

$$NDVIvar = NDVIpost - NDVIpre \qquad \text{where } NDVI = \frac{(B8-B4)}{(B8+B4)} \quad (2)$$

2. Variation of Modification of Normalized Difference Water Index (MDWI – equation 3). We use a similar index already
tested for MODIS data. We used Sentinel-2 to calculate the MNDWI considering the red edge band (B5) and the SWIR band
(B11) at 20 m of spatial resolution. With this approach, it is possible to map the variation soil moisture related to flooded areas
or areas that are still flooded.

$$MNDWIvar = MNDWIpost - MNDWIpre \qquad \text{where } MNDWI = \frac{(B5-B11)}{(B5+B11)} \quad (3)$$

## 3.2 Flood mapping at local scale with high and ultra-high resolution data

The flood mapping at local scale was made using high and ultra-high resolution images.
Immediately after the event, a research project proposed by CNR-IRPI was conducted with the participation of ALTEC S.p.A.,
Digisky s.r.l. and the Civil Protection Agency of the Metropolitan City of Turin. The aim of the project is a methodological
analysis of a possible low cost solution that could be used for high-resolution mapping of flood effects.
The study started from SMAT F2 Project results (Farfaglia et al., 2015), where different solutions for the acquisition of RGB
datasets with small and medium Remotely Piloted Aerial Systems (RPAS) were developed. Also previous experiences of CNR
IRPI and Civil Protection Agency in the use of small RPAS for the study of geo-hydrological processes (Giordan et al., 2015;
Boccardo et al 2015; Fiorucci et al., 2017; Giordan et al., 2017) were useful for the definition of a correct use of these systems.
These previous experiences pointed out how the use of low cost systems for the acquisition of RGB images and the application
of structure from motion algorithm (SfM) can be considered a good solution for the creation of high-resolution 3D models.

### 3.2.1 Aerial high-resolution images

Aerial photo took few hours or within few days after the peak of inundation allow mapping the flooded area with high precision
over the most involved territories. In our case, aerial photos were acquired after the flood over the Po river near the village of
Pancalieri and for Moncalieri town. We used a low cost aerial platform (Tecnam P92-JS) provided by DigiSky srl equipped
with Panasonic Lumix GX7 camera (mirrorless with 16 Mp) that allowed to acquire aerial photo with a spatial resolution of





10 cm/pixel. The system has also an on board GPS that acquires images shooting points and allows the georeferencing of the photos sequence using SfM.

The use of manned solution has several add values that can be very useful in this phase: i) it is possible to fly over urban areas without strong limitations that characterized RPAS, ii) the system is able to acquire images over large areas in limited lapse of time, iii) the system can flight on demand during the flood of immediately after (with favourable weather conditions).

The adopted solution was used to acquire 9.2 km$^2$ of the most damaged area of Moncalieri and 9.5 Km$^2$ of the flooded area of Pancalieri. These two areas are representative of different conditions:

1) The area of Pancalieri is a rural area mainly dedicated to the agriculture. In this case, the Po river flood covered large uninhabited sectors of the Pancalieri plain and reached part of the town of Pancalieri. Here, using SfM, the images of the plane were also processed for the creation of a DSM (resolution of 20 cm) with the aim of mapping geomorphological features

related to the flood.

2) The selected area of Moncalieri is a strong urbanized sector of the municipality. In this area there are: i) a motorway, ii) several regional and local streets iii) a residential area with recent unfamiliar houses and small condominiums, iv) an industrial and commercial district. The inundation of this area is due to the Chisola levees breaks. The map of Moncalieri was useful for the identification of most damages elements and in particular the levee. On the most critical areas, we used also RPAS to

acquire ultra-high resolution images.

### 3.2.2 RPAS ultra-high resolution images

The ultra-high resolution phase is based on the use of RPAS and a terrestrial system. RPAS were used to acquire nadiral photo sequences of the most damaged areas and infrastructures. In particular, we tested the possibility to use RPAS for the

identification of damages occurred to the Chisola river levee and one of the most damaged sectors of Moncalieri town. The employed RPAS is a multirotor (CarbonCore 950 octocopter) equipped with a Canon EOS M (Sensor CMOS APS-C, 18Mp). The system is equipped with a flight terminator and a parachute and can be used also in inhabited areas. The RPAS was provided by Civil Protection Service of Turin metropolitan area. The obtained aerial photos have a spatial resolution of 3 cm/Pixel. Using SfM, the images of the drone were also used to create 10 cm resolution DSM. All the flood mapping

methodology described until now are very often not able to give a consistent measure of water depth. This limitation is not due to the resolution or the time of acquisition, but it is intrinsic in nadiral images. For this reason, in Moncalieri area we choose to deploy a ground-based solution.



### 3.2.3 Ground-based ultra-high resolution images

As mentioned before, we tested a terrestrial low cost solution system for the acquisition of ultra-high resolution images. In particular, we used an integrated system developed by ALTEC SpA, which couple a Go-Pro HERO 3+ (Black Edition) camera with a GPS and an acquisition module. The system is able to record a STANAG 4609 geolocated HD video. The experimental system was installed over a CNR IRPI car and a survey was made few days after the flood in the considered area of Moncalieri. The continuous record of geolocated video can be a good solution for the acquisition of a large amount of data immediately

after the flood when marks of the water level are still clearly visible along building facades. The identification of water level of flooded areas based on the measurement of marks over facades is not a novelty, but the manual acquisition of these data has often been a critical task. Citizens often want to quickly obliterate these signs as a reaction to the critical experience that they lived. The use of field teams that look for these marks can be a time consuming task that can produce few results with large efforts also because before the survey it is very difficult to have an idea of the number and the distribution of facades marks

that can be identified and measured. The number of marks strongly decreased after few weeks and for this reason, it is important to have a system that can acquire very fast geolocated images and that can be easily used over large areas.

The presented system is very simple and effective. The geolocated video can be analysed by other components of the team immediately after the acquisition or after many days. The important goal is the fast acquisition of numerical information of the flood effects that can be used for several purposes. For the identification and mapping of water levels, the video is analysed

and a frame sequence is extracted from it when the operator sees some marks lefts by water over facades. The developed system is able to extract not only frames but also their geocoding information, which are computed using SfM applications. The result is a georeferenced 3D model of the façade that can be used to measure the water level with a good approximation (few cm).

### 3.2.4 Field data

Field survey, ancillary data like a measure of groundwater discharge stations, and civil protection reports were used to validate the maps derived from remote sensing interpretation and the simulation models for Pancalieri and Moncalieri areas. We made a GPS RTK campaign in Tetti Piatti and Tagliaferro areas to have direct measurements of flood marks. In particular, we acquired the 3D position of marks previously identified using the available video. We used third part materials like newspaper

reports, photo and videos found on the web with a validation of their reliability in terms of geolocation and time. Available data were used to check the extension of flooded area and water height mapped with other methodologies.





### 3.3 Water depth models based on DTM approach.

As mentioned before, the main goal of the presented methodology is the definition of the maximum depth reached by the water during the flood. The map of water maximum depth (WD) is an important document that can be used for a first definition of damages and remedial actions. All the acquired material, and in particular data that define the water depth reached by the flood, were used to calculate the water maximum depth map. In our study, we adopted a simple raster-based model (Bates and De Roo, 2000) and we created a water absolute level (WL) raster. The first step is the definition of WL is the acquisition of several measure points of water level that are provided by the estimation of water depth ($WD_p$). WD measures can be done using: i) georeferenced photos (low accuracy), ii) ultra-high resolution phase results (high accuracy), iii) direct measurements supported by GPS RTK positioning, iv) civil protections reports (the level of accuracy can be very different), v) data acquired by hydrometric river level monitoring stations. Starting from the collected punctual measures and the 5-m LIDAR DTM freely provided by Regione Piemonte, we calculated the WL value for each point using a simple formula: $WL_p = DTM + WD_p$. Available WL points were used to create the water level contour lines and then interpolated using GIS software to obtain the raster of WL gradient. The WL raster is used to create the raster map of water depth which can be calculated with a simple raster calculator of a GIS software using the reversed formula WD=WL-DTM. WD maps is an important information to assesses and improve the limit of the flooded area it is also fundamental for the phase of preliminary damages assessment. We produced maps of water depth at medium resolution for Po river (Fig. 5) and at high resolution for Moncalieri (Fig. 11) and Pancalieri area (Fig 6).

### 4 Results

### 4.1 Flood mapping from low to medium-high resolutions with satellite data.

The satellite data allowed to map the flooded areas by Po river, Chisolm and Oitana stream, with a resolution that ranges from 500 m of MODIS to 10 m of Sentinel-2. In following figures (Fig.3. Fig 4. and Fig.5), remote sensing maps are compared with our limit of the flooded area (black polygon). The flooded area limits were manually extrapolated considering satellite data and geomorphological features obtained using the hillshade model derived from 5-m DTM of Regione Piemonte and used as a benchmark for the evaluation of the performance of remote sensing analyses. For Po and part of Chisola, the flooded areas were mapped also with help of water height simulation on the base of DTM. At the moment of writing this paper (November 2017), it is still not available an official delimitation of flooded areas, a map made by ARPA Piemonte is under validation and the data will be published in the next months.





### 4.1.1 Flood mapping with SAR data

I) Co-flood mapping with COSMO-SkyMed data. Results of data classification are reported in Figure 3A, where four main classes, i.e., water/flooded (blue), soil/vegetation (green), urban (pink) and quarry lake from ancillary data (cyan), are shown. The analysis could not detect all flooded areas because the COSMO-SkyMed data was acquired in the early morning of 25 November while the phase of the maximum flood of Po river occurred on 25 November afternoon. At the time SAR data was acquired (05:05 UTC), the Pancalieri area was at initial stage of flooding and the urban area of Moncalieri was not flooded yet (the flooding in this area started a few hours later). Only the Oitana stream already flooded over most of the areas. It is important to note that, in the COSMO-SkyMed image, some areas classified as water are not flooded zones but quarry lakes. Accordingly, the analysis based on COSMO-SkyMed data cannot provide an exhaustive flood map but results may be considered satisfactory in terms of spatial distribution of flooded areas.

II) Post-flood mapping with Sentinel-1 data. Figure 3B shows the map of the post- pre-flood SAR backscatter difference ($\Delta\sigma^o_{post-pre-flooding}$) where the application of empiric thresholds allowed us detecting areas covered by water, i.e., flooded ($\Delta\sigma^o <$ - 1dB). Such results show that most of the areas classified as flooded by the co-flood analysis were not anymore covered by water on 28 November 2016. Only small depressed areas, e.g., ancient meanders of Po river, were still flooded, as shown in Figures 3D', 3D'' and 3D''' (the area of Pancalieri, discussed more deeply in par 4.2.).

### 4.1.2 Flood mapping with multispectral data

**I) Multispectral low resolution, MODIS-Aqua**. The MODIS-Aqua satellite takes an image quite free of clouds over the entire Piemonte during the late morning of November 26, 2016. The image allowed to detect the flooded areas with a resolution of 500 m.

From the false colour images (Fig. 4 B), even if the area at south of Turin is not yet directly flooded, it is only possible to detect that the soil was saturated of water (dark green-blue in false colour composition). The identification of flooded area is more evident from the comparison with pre-flood image of November 12, 2016 (Fig. 4 A).

We also try to extract in an automatic way the flooded area:

In figure 4 C we identified flooded area using the value MNDWI$_{var}$>0.3. This is an empirical threshold that minimizes errors.

The results show a good correspondence between the area manually drawn and the area automatically classified even if around 35% of the flooded area was not correctly identified. This can be explained if the satellite passed at the end of the co-flood stage. It is also possible to see some false positive pixels that correspond to the shadow of the clouds even if clouds were partially filtered.





In figure 4 D shows the maximum likelihood supervised classification of co-flood image. In the study area, we classified four
main land cover: vegetation, bare soil, cloud, and water / wet soil that should identify the flooded area. Supervised classification
identified most of the flooded areas (> 80 %) but with more false positive cases that correspond to cloud shadow areas classified
as water.

For both indexes, it possible to see that the area of Moncalieri (red square 1 in figure 3) flooded by Chisola stream is not well
identified.

**II) Multispectral medium-high resolution post-flood mapping Sentinel-2.** The images of Sentinel-2 were analysed by
visual interpretation of RGB composite image and using two different indexes (NDVI – MNDVI) to identify flooded areas
shown in figure 5.

1) NDVI variation ($NDVI_{VAR}$) at 10 m of Spatial resolution (Fig. 5 A). The results show that for Po, Chisola and Oitana a clear
pattern of negative NDVI variation corresponds to the flooded area. The study area is almost flat and mostly occupied by
cultivated fields and, in November, it was characterized by tillering of wheat. The flood caused a deposition of a thin layer of
silt sediment that caused a decrease of vegetation activity (most of the flooded area shows an $NDVI_{VAR}$ <-0.06) that could be
detected using the available dataset. By the contrary, the wheat field outside flooded area shows an increasing or stationary
NDVI. In the maps are visible negative $NDVI_{VAR}$ also outside the flooded area that is related to: i) winter decreasing of activity
of natural vegetation or some type of cultivations, ii) longest building shadow in urban areas. The presence of false positives
hampers the use of automatic classifications of flooded areas and a visual interpretation is necessary. It is possible that flood
effects and the layers of silts could have affected also the crop productivity with relative economic damages as reported in
other cases (Tapia-Silva et al., 2011; Shrestha et al., 2017), but this evaluation is not the aim of this study.

2) MNDWI variation ($MNDWI_{VAR}$) at 20 m of spatial resolution. The index is directly related to the presence of water or high
soil moisture. The results show that for Po, Chisola and Oitana areas (Fig. 5 B) a clear pattern of positive $MNDWI_{VAR}$ that
indicates an increase of soil moisture and the presence of some areas that were still inundated. It is possible to see that a
threshold of $MNDWI_{VAR}$ + 0.1 is the best to delimit the flooded areas. However, like for NDVI, the presence of many areas
with positive variations outside the flooded sector makes more accurate a manual interpretation with respect to an automatic
classification. The evidence also suggests that for this index it is important to have images taken within few days after the
flood when the involved areas are still flooded or very water saturated. It is worth to note that for Sentinel-2 for both
$MNDWI_{VAR}$ and $NDVI_{VAR}$ the flooded area is well detectable in the local area of Pancalieri where land use is mostly cultivated
land, while is more difficult to detect the flooded are in the case of Moncalieri that is mostly urbanised.





**III) Water depth model.** The water depth model for Po river was created following the procedure described in paragraph 3.3,
the results are shown in figure 5C. The simulated WD model has a very good match with benchmark polygon and the evidence
from Sentinel-2 (Fig. 5 A and B), MODIS data (Fig. 4). It is also possible to observe some discrepancy at South-East of
Moncalieri where large area should be flooded according to model but in reality, was not affected. This mismatch could be
explained by the presence of artificial structures (e.g. embankment) that protect flood prone areas and cannot be simulated in
our model. Over this large area we have not ground measure for validation but it is possible to estimate from some photos
found on the web that model error is within 0.5 m. In the higher resolution WD model of Pancalieri and Moncalieri shows in
the next chapter was possible to validate data with ground truth evidence.

The final limits of the flooded area are the results of both remote sensing and WD model interpretation its accuracy can be
considered good for cultivated area and large flooded area by Po river but less accurate for urban area especially in Moncalieri
where a local high-resolution analysis is needed to quantify the severity of the flood.

## 4.2 Flood mapping at local scale with high resolution data.

Inside the area analysed using remote sensing systems, we choose the most critical sectors of Moncalieri and Pancaleri to test
high and ultra-high resolution images. As mentioned before, the high resolution has been acquired using an aircraft, and the
ultra-high resolution using RPASs and a ground-based photo system. All the images have processed using SfM that allowed
to obtain ortophoto and 3D models.

### 4.2.1 High-resolution aerial photo - Pancalieri

The village of Pancalieri was partly flooded by Po river in the morning of November 25, 2016. This area has been mapped
also by high resolution aerial photo (10 cm/Pixel) in visible bands provided by DigiSky. Aerial photos were taken November
28, 2016 (figure 6 A) and allowed to refine the map of the flooded area form the medium resolution maps obtained with the
interpretation of Sentinel-2 data. With the help of digital surface models (DSM) at 0.2 m of spatial resolution derived from
SfM, we also mappe7d geomorphological features like erosion (meanders cut) and deposition areas and road damages (Fig. 7
C). The integration of aerial photo and DSM allowed also to make a 3-D model where it was possible to measure water depth
for some points where water level marks are well detectable (Fig. 7 B).

Using the procedure described in paragraph 3.3 we produced a WD model for the Pancalieri area (Fig. 6 D) with higher
accuracy respect to the rest of Po valley. The higher accuracy of the model was obtained using: i) high resolution aerial photos
ii) spot measures derived from SfM iii) different video and photos found on the web and geolocalized with the help of Google





Street view (e.g. Fig. 6 B and 6C). The model shows that part of the Pancalieri village was flooded by a modest height of water (< 0.5 m) while near Po river WD reached 2-3 m with a strong flow that caused erosion channel. From the map of the flooded area (Fig. 6 D) it is also interesting to note that, during the flood, ancient meanders at the east of Pancalieri were reactivated

and ad consequence some areas were flooded quite far from Po main course.

Some months after the flood (April 2017), satellite photos available on Google Earth (0.5 m spatial resolution) still show some trace of flood like erosions and area covered by sand deposits. The flooded area is much more difficult to identify and confirm the importance to acquire data as soon as possible after a flood event.

### 4.2.2 High resolution aerial photo and ultra-high resolution RPAS 3D models - Moncalieri.

Some parts of Moncalieri municipality (Tetti Piatti, Carpice and Tagliaferro localities) were flooded in the late morning of 25 November by Chisola stream that breached its embankment in different points (Fig. 8 B). On the left side of Chisola, the area with a very dense residential and industrial settlement suffered strong damages. Hundreds of people were evacuated. In the evening of November 25, another sector of Moncalieri municipality was flooded by Po river.

Few days after the flood, the area flooded by Chisola was analysed with different methodologies:

1. High resolution aerial photos. Like in Pancalieri, on November 29 a very high resolution (0.1 m/pixel) aerial photo using the aerial platform of DigiSKY was taken over an area of about 9.2 km$^2$. The aerial photo allowed to refine the map of flooded areas (Fig. 8 A) and to detect the points where river embankment collapsed (Fig. 8 B).

2. Ultra-high resolution RPASs photos. On December 3, 2016, RPAS photos (resolution of 0.02 / 0.03 m/pixel) were acquired over some most critical areas (e.g. Tetti Piatti – Fig. 8 C) for precise a mapping of flood effects. In this area, ultra-high

resolution allowed to detect some damages like the toppling of a wall in recently built urbanization or the waste accumulation derived from damaged objects originally located in houses or industrial warehouse. The presence of these deposits is a clear evidence of the occurred damages, but also a confirmation that nadiral images are not able to supply a sufficient dataset for the identification and evaluation of damages in urban areas.

The DSM based on RPAS photos allowed also to create a detailed 3-D model of river embankment rupture (Fig. 9). The

presented 3D model confirmed that the level of Chisola during the flood was very critical with a difference of fewer than 0.5 m with respect to the top of the levee. The water maximum level considered under the security limits suggested by Po river authority is 1 m with respect to the top of the embankment. The RPAS model allowed also to map geomorphological effects of the rupture of the embankment. In particular, figure 8B and figure 9 show a strong erosion near the break and a pseudo alluvial fan created by the flow of water.





### 4.2.3 Measure of water depth with SfM model from terrestrial camera Moncalieri.

In the same days of UAV an aerial photo campaigns, a field survey using an integrated system provided by ALTEC S.p.A. installed on a car (Fig. 10 B) was made in the same urban areas of Moncalieri flooded by Chisola (Fig 10A). The survey had the aim of measuring the maximum water level reached and a rapid evaluation of damages. The survey last about 1h for 12 kms of the path along the road of the most critical area hit by the flood.

Where the level reached by the water was still visible over several facades, (Fig. 10 C and Fig.10 D) it was possible to make an estimation of the maximum water level of the flood. During the first survey, we found 11 points where watermarks over facades were still visible. During the post processing, we realized that the quality of the images extracted from the video was insufficient for the SfM application. For this reason, after a month we performed a second survey along the same path, but only 6 marks still visible (Fig 10 A). This reduction of available points confirmed that the delay between the flood and the survey is a fundamental element that should be carefully considered because the number of possible information decrease exponentially. For this second terrestrial camera acquisition, an improvement of the encoding quality was introduced. Such improvement allowed the extraction of good quality images compatible with SfM application. We obtained 3D models of the surveyed sectors and we measured the high of water marks on façades. Then we validated the information obtained from SfM with a manual water height measurement geocoded with GPS RtK systems for the 6 points and for other additional 5 points. The accuracy of measurement considering that is a low-cost solution and one of the first experimental tries for this system is very good: the average error compared SfM water level measurement with manual measure can be estimated in few centimetres (see table 6).

### 4.2.4 High resolution water depth models and ancillary data for damages evaluations - Moncalieri.

The combined use of measures derived from: i) car camera elaborated with SfM, ii) manual GPS RTK, iii) the hydrometric level of Chisola stream registered by ARPA Piemonte station (Fig. 11 B) represents a good dataset for the estimation of the WD. Using the 5m DTM Lidar of Regione Piemonte, we obtained the WL and the WD rasters (Fig. 11 A). The result shows that in a large part of the analysed area, the water height was between 0.5 and 1 m. Unfortunately, in some morphological depressions, the level was higher than 1.5 m. The model shows also that in the cultivated area close to left Chisola embankment water probably reached 2 -3 m height.

The water level map can suffer from some errors from punctual measure these are related to the quality of DTM, or the effect of local structures that can modify the water flow and height at local scale. The comparison of water level measured with SfM / GPS and calculated level with DTM show variation within 0.2 m that is a good result (Table 6).





Ancillary data like photos or video found on the web (local newspaper, social media) and geolocated with Google Streetview
allowed to improve and validate the map of flooded areas and the height of the water (Fig 11 C). On the web, it is possible to
find a lot of photos or video of flood event but only small part of them can be geolocated with good precision and validated.

The water height map was crossed with buildings database of Regione Piemonte to assign to each building the average of
water height reached by the flood (Fig 12 A). The water height is one of the main parameters that can be used for a prelimanry
estimation of potential damages. We dived the water height in 3 main classes corresponding to low (<0.5 m), medium (0.5 –
1.5) and high (> 1.5) damages expected. These thresholds have been empirically defined by Luino et al. (2009) and Amadio
et al (2016). The obtained map is a good representation of the level of damages caused by the flood that could be considered
the final product of the presented methodology.

This result was compared with ground data where possible: for instance, in the industrial warehouse shows in detail map in
figure 12 B was estimated an average value of 0.8 m water level (medium degree of damage expected). The evidence from a
geolocated photo from La Stampa newspaper confirms this value.

## 5 Discussion and conclusions

In this work, we tested different methodologies for a low cost and rapid flood mapping and characterization using the
November 2016 Piemonte flood as a case history. We used a multiscale and multi-sensors approach in order to know pros and
cons of each methodology in relation to the site conditions and available data

At regional scale, satellite remote sensing showed a good performance in the flood mapping: the combined used of InSAR
data of Sentinel-1 and Cosmo, and multispectral data of MODIS-Aqua and Sentinel-2 allowed creating maps of the flooded
area.

InSAR data showed a good performance in the real time flood mapping while are weaker for post event mapping. By
considering the obtained results it is also clear the importance to have free and constant SAR satellite data provided by national
agency: a short revisit time and a constant acquisition are factors that increase the probability to have SAR image for real-time
flood mapping. For instance, the two Sentinel-1 provide free images every 6 days, while other satellites have quite high costs
and the acquisition of the extra images is activated with emergency procedure acquisition like the EMSR of the European
Union or by civil protection authorities that not always provide useful maps for whole damages assessment.

Multi-spectral data have more capability with post event images. In this work, we tested NDVI and MNDWI variations for the
detection of flooded areas based on the comparison of pre and post event images. Both methodologies show good-performance





in cultivated land, while is more difficult the interpretation of images for dense urban areas. In last years, the revisit time for free multispectral data has been strongly reduced (Landsat-8 has a revisit time of 16-days and Sentinel-2 f 5 days from march 2017 with the launch 2$^{nd}$ satellite). This increase the probability to have an image free of cloud within few days or weeks after

the flood or in some cases during the inundation phase.

At local scale, flood mapping showed a good agreement with regional scale mapping.

The high resolution aerial photo and ultra-high resolution aerial photo from RPAS allowed to map flooded areas with more precision. The application of Structure from Motion (SfM) allowed creating high resolution DSM useful to map the geomorphological effects (e.g. meanders cut) and the main damages (embankment rupture) in Pancalieri and Moncalieri area.

For the considered urban area (Moncalieri), where satellite data is weaker and a precise evaluation of water height is very important for flood damages evaluation, the solution is the acquisition of ground-based data. In our work, we tested a low cost solution with a GO-PRO HERO 3+ (Black Edition) camera installed on a car that allowed to make 3D models and to measure the water height reached during the flood. These measures validated with GPS showed a good accuracy.

A possible idea is to use this system during the emergencies, for instance, on civil protection car, in order to have a map of

water height with much higher density of points.

Using these measures and a high resolution DTM, it was possible to generate a raster model of water depth that has a good match with ground truth and could be used for the evaluation of building damages or for flood prevention policy. This model can be also used for the estimation of the degree of damage for every building located in the flooded area.

Finally, from our work it is also clear the importance to collect ancillary data also from the new sources on the web: the photo

and video collected during the flood by simple citizens can be a precious help for the validation of flooded area maps.

**Data availability.**

MODIS data were downloaded from Worldview portal (https://worldview.earthdata.nasa.gov/?) link retrieved 21-11-2017
12/11/2016: MYD09GA.A2016317.h18v04.006.2016319100800 view: https://go.nasa.gov/2ymajYo
26/11/2016: MYD09GA.A2016331.h18v04.006.2016333055328 view https://go.nasa.gov/2y4keBY


Sentinel data   were download from Copernicus Scihub: https://scihub.copernicus.eu/  link retrieved 21-11-2017
Sentinel-2:
1/12/2016: S2A_OPER_MSI_L1C_TL_SGS__20161201T104644_20161201T141912_A007541_T32TLQ_N02_04_01
8/11/2016: S2A_OPER_MSI_L1C_TL_SGS__20161108T103641_20161108T154744_A007212_T32TMQ_N02_04_01

Sentinel-1:



28/11/2016: S1A_IW_GRDH_1SDV_20161128T053526_20161128T053551_014138_016D3F_7094

22/11/2016: S1B_IW_GRDH_1SDV_20161122T053445_20161122T053514_003067_005376_AD1

Cosmo-Skymed quiklook image (E-Geos http://www.e-geos.it/)

ID_627100 acquired on 25 November   COSMO-SkyMed© ASI [2016] http://catalog.e-geos.it/#product:productIds=627100
link retrieved 16-11-2017

 5-m LIDAR  DTM Regione Piemonte is available at:

http://www.geoportale.piemonte.it/geonetworkrp/srv/ita/metadata.show?id=2552&currTab=rndt   link retrieved 21-11-2017


**Acknowledgments.**

The authors gratefully acknowledge Italian Space Agency (ASI) and E-GEOS  for the permission of free use of. Cosmo-Skymed quick-look data.  ARPA and Regione Piemonte for the support in the search of meteorological and hydrological data.

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



**Table 1. Resume of dataset used to map and characterize flooded area in this study**

| Type | Sensor used | Spatial resolution (m) | Covered area by single scene (km²) | Min. Revisit time (Day) |
|---|---|---|---|---|
| 1 – Satellite Data | | | | |
| SAR –X band | Cosmo-SkyMed | 60 | > 1'000 | 4 |
| SAR- C band | Sentinel-1A/B | 5 (ground range) x 20 (azimuth) | > 10'000 | 6 |
| Multi-spectral | MODIS-Aqua | 500 | > 100'0000 | Daily |
| Multi-spectral | Sentinel-2 | 10 / 20 | > 10'000 | 10 (5) |
| 2- Aerial data | | | | |
| Very High res. visible band | Tecnam P92-JS | 0.01 | 100 km² | On demand |
| Ultra-High resolution visible band | RPASs CarbonCore 950 octocopter | 0.02 / 0.03 | < 10 km² | On demand |
| DTM LIDAR | Airborne | 5 | Piemonte region | Archive data |
| 3- Ground-Based | | | | |
| Photo / video from car platform | GO-PRO HERO 3+ | 0.02 / 0.03 | Local /urban | On demand |


**Table 2. Resume of satellite data in relation with flood stage**

| Satellite | Spatial Resolution | Acquisition time | | |
|---|---|---|---|---|
| | | Pre-flood | Co-flood | Post-flood |
| Cosmo-SkyMed | Medium | | 05:05 UTC – 25/11/2016 | |
| Sentinel-1 A/B | Medium | 05:35 UTC – 22/11/2016 | | 05:35 UTC – 28/11/2016 |
| MODIS Aqua | Medium-Low | 12:30 UTC - 12/11/2016 | 12:30 UTC - 26/11/2016 | |
| Sentinel-2 | Medium-High | 15:19 UTC - 11/11/2016 | | 14:19 UTC – 01/12/2016 |






**Table 3. Characteristics of the Sentinel-1 dataset used in this study**

| | |
|---|---|
| Satellite | Sentinel-1 A/B |
| Sensor Parameter | C-band 5.405 GHz |
| Orbit | Descending |
| Pre-flood acquisitions | 22/11/2016 |
| Post-flood acquisitions | 28/11/2016 |
| Data format | Single Look Complex (SLC) |
| Azimuth pixel spacing [m] | ~13 |
| Range pixel spacing [m] | ~2 |


**Table 4. Characteristics of the Aqua MODIS data used in this work.**

| Band | Bandwidth nm | Band type | Spatial resolution (m) |
|---|---|---|---|
| B1 | 620 – 670 | Red | 500 |
| B2 | 841 – 876 | NIR | 500 |
| B3 | 459 – 479 | Blue | 500 |
| B4 | 545 – 565 | Green | 500 |
| B5 | 1230 – 1250 | SWIR | 500 |
| B7 | 2105 – 2155 | SWIR | 500 |

**Table 5. Characteristics of Sentinel-2 data used in this work.**

| Band | Wavelength nm | Band type | Spatial resolution (m) |
|---|---|---|---|
| B2 | 490 | Blue | 10 |
| B3 | 560 | Green | 10 |
| B4 | 665 | Red | 10 |
| B8 | 842 | NIR | 10 |
| B5 | 705 | NIR | 20 |
| B6 | 740 | NIR | 20 |
| B7 | 783 | NIR | 20 |
| B8a | 865 | NIR | 20 |
| B11 | 1610 | SWIR | 20 |
| B12 | 2190 | SWIR | 20 |






**Table 6. Water height measures obtained from structure from motion, GPS survey and simulation with DTM**

| Measure point coordinates | | Water Depth (m) | | |
|---|---|---|---|---|
| UTM X | UTM Y | SfM (+/- 0.05) | GPS | DTM (+/- 0.2) |
| 395132 | 4983240 | 1.56 | 1.60 | 1.61 |
| 395242 | 4983152 | 1.45 | 1.40 | 1.42 |
| 395140 | 4982644 | 0.84 | 0.78 | 1.01 |
| 395142 | 4981624 | 0.82 | 0.81 | 0.78 |
| 395022 | 4981188 | 1.28 | 1.35 | 1.56 |
| 394877 | 4980993 | 1.40 | 1.37 | 1.34 |






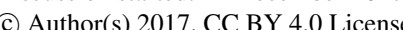

**Figure 1: A) Location of Piemonte region in Italy, B); B) Rainfall in Piemonte region during the 21-25 November flood event (Based on ARPA Piemonte data) and location of study area, S. B. = Savigliano Basin, P. P. = Poirino Plateau, T.H = Turin Hills; C) Detailed view of study area with discharge in the stream gauge stations and the location of Pancalieri (1) and Moncalieri (2) case History.**




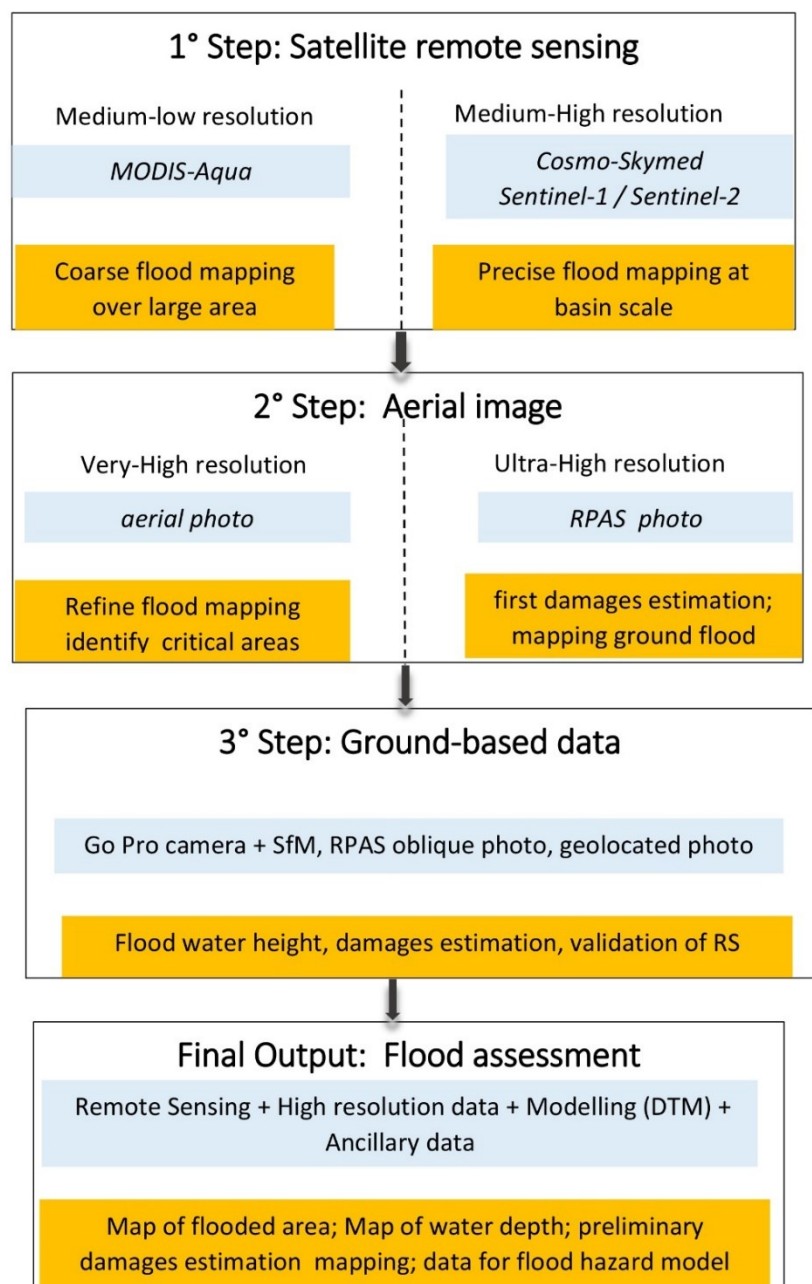

**Figure 2: The flowchart illustrating the multiscale flood mapping approaches proposed in this work**




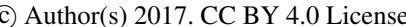

**Figure 3: A) Reclassified Quicklook Amplitude SAR Image acquired at 05:05 UTC of 25 November 2016 - COSMO-SkyMed© ASI [2016] - B) Sentinel-1 geocoded backscattering coefficient difference (Δσ°); C) Example of change backscattering between the pre- and post- flood image is clearly detectable. D'; D'' D''') detail of some areas where Sentinel-1 still detect water.**




**Figure 4: MODIS Aqua satellite image A) False colour band composition 7-2-1 acquired the 12 November 2016.; B) False colour**

**band composition 7-2-1 acquired the 26 November 2016.; C) automatic detection of the flooded area using** $MNDWI_{var} \gg 0.3$; **C)**

**automatic detection of the flooded area using supervised maximum likelihood classification. The red box identifies the local case**

**history of Moncaleri (1) and Pancalieri (2)**



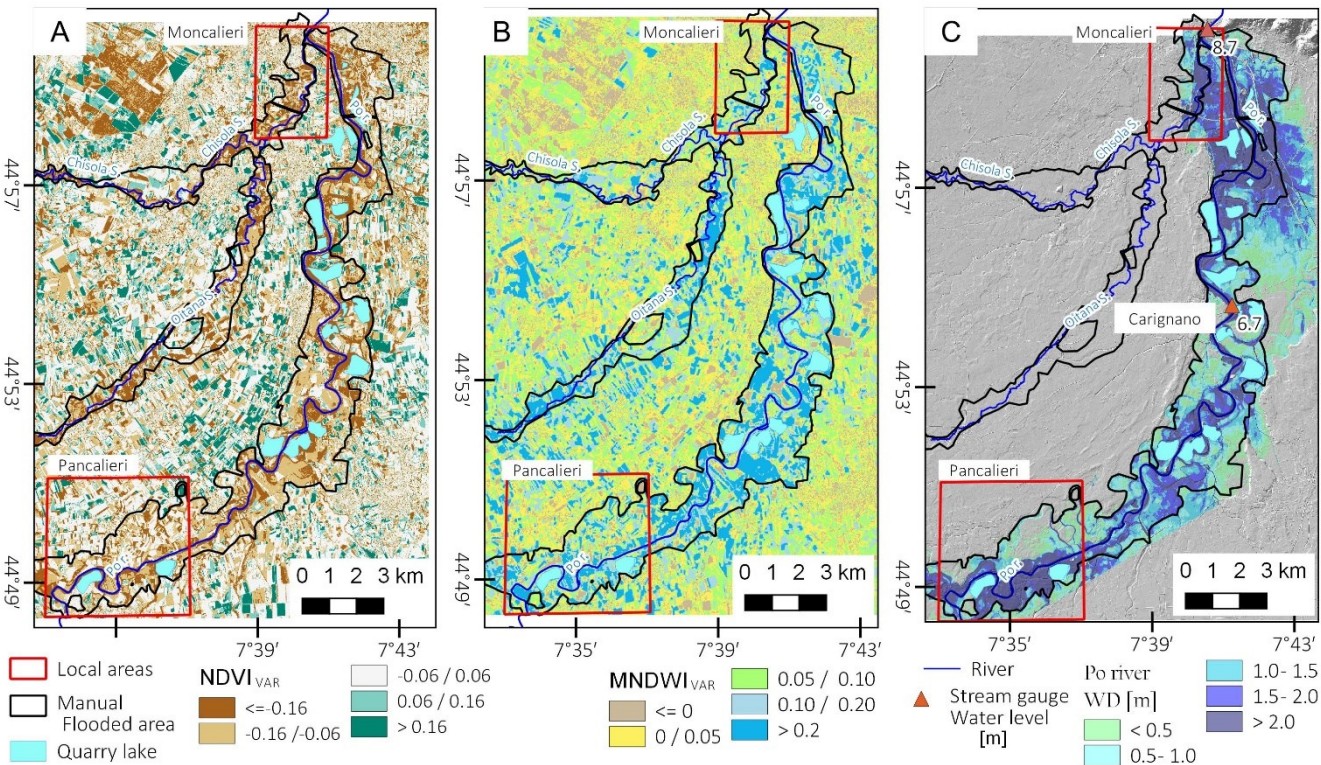

**Figure 5: Sentinel-2 analysis and validation: A) NDVI variation 10 m of spatial resolution, B) MNDWI variation 20 m of spatial resolution, C) Simulation of flooded area water depth for Po river based on 5-m DTM and river height level registered in Arpa Piemonte stream gauge.**



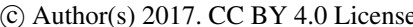

**Figure 6: A) Aerial photo at 10 cm of resolution taken the 28 November 2016; B and C) photo took from a local newspaper and geolocalized with Google Streetview; D) Geomorphological elements and model of estimated water depth (m).**




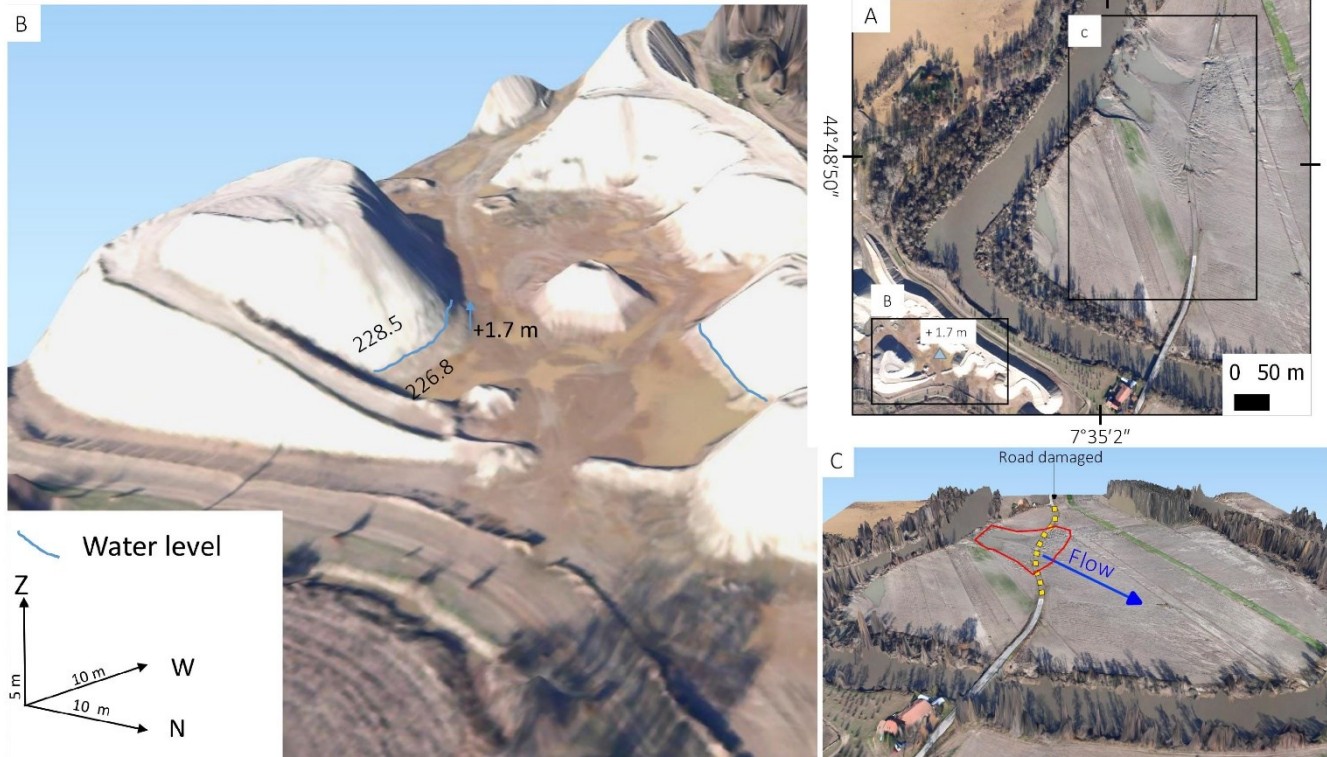

**Figure 7: The SfM 3-D model obtained from very high resolution aerial photo allowed to measure the approximate height on water on a sand deposits of a quarry (B) near Po River at south of Pancalieri (A) and to observe the effect of meander cut: an erosion channel and the destruction along a stretch of road (C).**






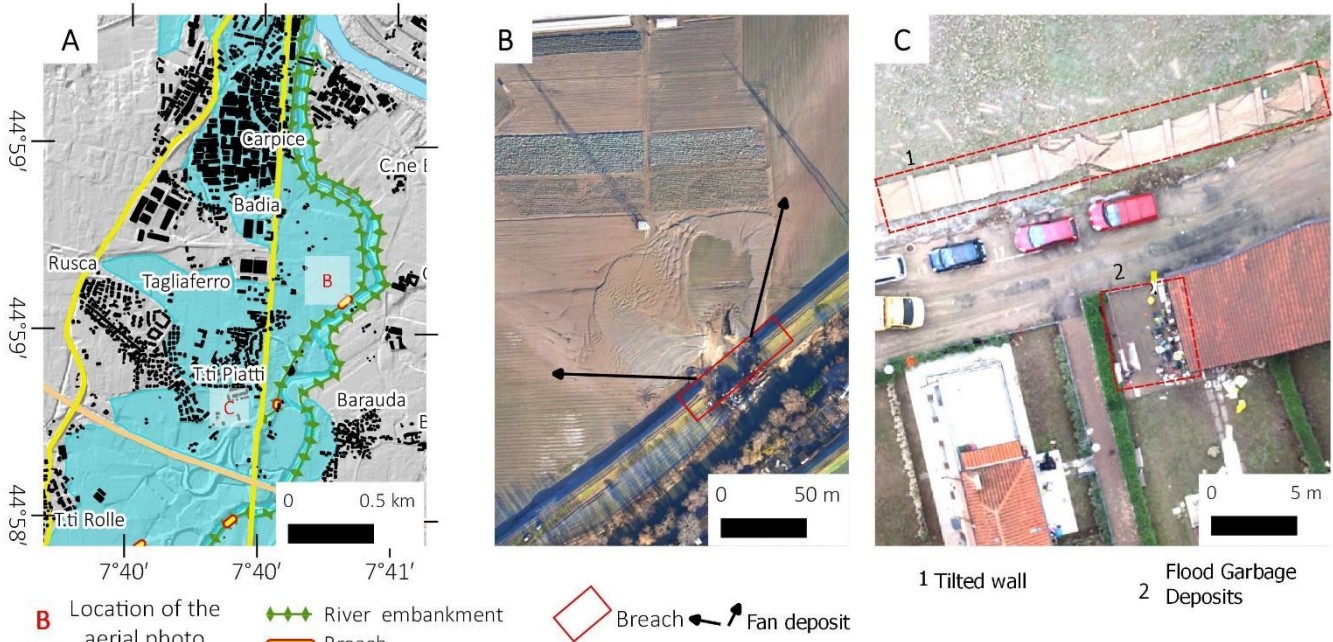

**Figure 8: map of flooded Chisola flooded area in Moncalieri municipality with location of detailed photo; B) Aerial photo at 0.1 m of resolution showing the breach of river embankment and the 'alluvial fan' created by water flow showed with 3-d view in figure 9; C) RPASs photo at 0.02 m of spatial resolution showing a collapsed wall in the Tetti Piatti area and the deposit of damaged good from house.**


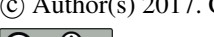


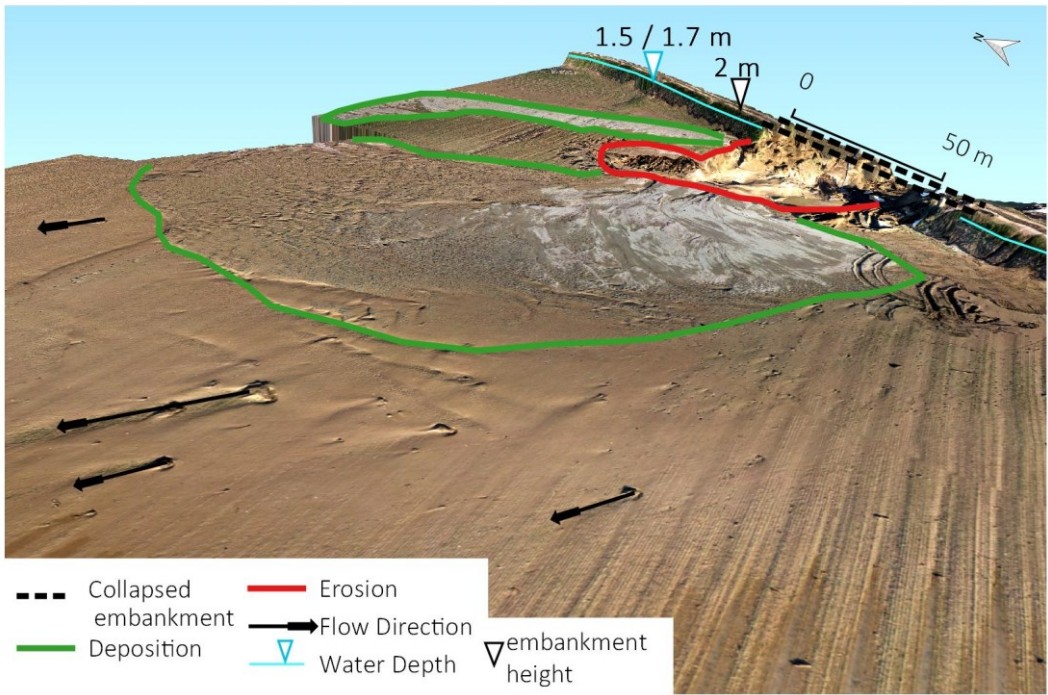

**Figure 9: 3-D Model derived from the RPASs aerial photo and SfM elaborations photo overlap. The model shows the river embankment rupture (point B in figure 8A), the geomorphological effects in the neighbour areas and allowed the estimation of water depth from the signs on embankment.**




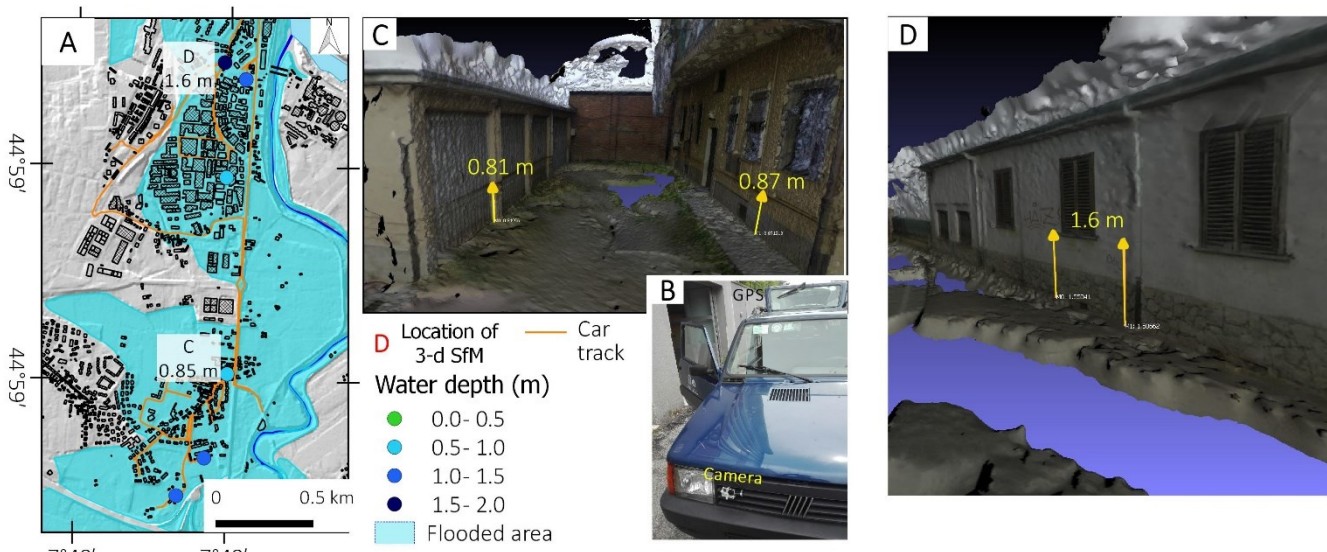

**Figure 10: A) Map of flooded area by Chisola stream in Moncalieri municipality with measured water height and location of 3-d**
**photo; B) Installation of the GO-PRO HERO 3+ (Black Edition) camera and GPS antenna over the car (the processing system for**
**STANAG 4609 encoding was installed inside the car); C and D ) Examples 3-d models made with Structure from Motion in which**
**was possible to measure the water height.**






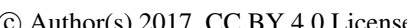

Figure 11: A) Water depth levee map based on SfM data and 5-m DTM model; B) Water level reached by Chisola stream in ARPA Piemonte station; C) Geolocated third part photo: 25/11/2016 aerial view of flooded area find on the web (https://vivere-moncalieri.it/2016/11/30/3760/ )



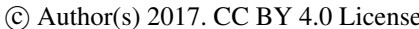



**Figure 12:A) Water level height map based on SfM data and 5-m DTM model; C) Zoom on the building of figure D; B and D)**

**Photo where it is possible to observe the water depth from the newspaper "La Stampa" geolocated using Google Street view.**