# Peer review of "Low cost, multiscale and multi-sensor application for flooded areas mapping."

_Natural Hazards and Earth System Sciences, 2017_

## Referee Comment (RC1) · Anonymous Referee #1 · 21 Jan 2018

The manuscript includes an original study on flood mapping using various remote sensing image sources and techniques. Therefore it has practical significance. In the literature, as also referenced in the study, there are so many research articles studying the evaluated data types and the techniques, however this study uses most of the available data sources and techniques for a single case showing the efficiency of results. Therefore, a comparative study in which the results of maps using optical and SAR images processed with different remote sensing techniques is presented. In general, the proposed approach was explained well, the experiments were conducted properly, and the results were discussed in the manuscript. However, there still exists some missing points in the manuscript in terms of the completeness of the paper. Therefore, if they are corrected considering the minor issues highlighted below, the article is
recommendable for publication.

Reviewer recommends: # In section 3.1.1 - It is noted that available COSMO-SkyMed image has been classified into three main land cover classes as; water-covered areas, i.e., flooded (low amplitude), urban areas (high amplitude) and soil/vegetation (intermediate amplitude). There, what is the type of the classification method used? The result of the accuracy assessment of the classification process was not given? Authors are recommended to give at least the overall accuracy of the classification! Please also note that in section 4.1.1, the classification accuracy of COSMO-SkyMed was not presented. # In Section 3.1.2 - Did the authors apply an atmospheric correction to MOS-DIS data? - It was also noted that a supervised classification was applied to MODIS by SAGA-GIS. Which supervised classification method was used? Quantify the accuracy of the classification result. # In Section 3.2.1 and 3.2.2 - DSM is generated from high resolution images. Digital Surface Model (DSM) is not a Digital Terrain Model (DTM). Authors should know the difference between surface and terrain model. # In section 3.3 - How did the authors perform water level measurements by GPS-RTK positioning? Give a little detail. # In section 4.1.2 - In Figure 4, in the figure caption, the letter of the final item D) appears as C) second time! Correct it. - It is observed that MODIS image is classified into Cloud, Water, Wet soil, Vegetation, and Bare soil whereas COSMO-SkyMed image has been classified into three main land cover classes as water-covered areas (i.e., flooded), urban areas, and soil/vegetation. It looks like only a GIS guery can be done between the classes "Water" and "Water-covered areas" classes derived from CSKM and MODIS images, respectively. Authors need to explain in detail how they used maps generated from the classified images. # In Section 4.2.3 - Authors declared that " During the post-processing, we realized that the quality of the images extracted from the video was insufficient for the SfM application. For this reason, after a month we performed a second survey along the same path" Explain the insufficient qualifications for the extracted images used for the SfM application. # In Section 5 - It was written that ......" the combined used of InSAR data of Sentinel-1 and Cosmo. and multispectral data of MODIS-Agua and Sentinel-2 allowed creating maps of the
flooded area. InSAR data showed a good performance in the real time flood mapping while are weaker for post event mapping......." Here, instead of InSAR data, the use of SAR data is recommended. It is because the only amplitude value of the SAR data was used and no interferometric process was applied. # In the discussion and results section - Rather than using expressions such as "good agreement", "more precision", "good accuracy", etc; quantify the accuracy or the quality of maps, results, etc.

Last but not least, the difficulty of this study is that the satellite data might have not always been received at the time of the hazard occurred! The authors can add a better flowchart that shows the missing data can be replaced by the other, taking into account the image data sorted from high resolution to low resolution.

Please also note the supplement to this comment: https://www.nat-hazards-earth-syst-sci-discuss.net/nhess-2017-420/nhess-2017-420-RC1-supplement.pdf

---

## Author Comment (AC1) · 6 Feb 2018

We would like to thank you the Reviewer for the detailed revision and his important suggestions. We improved the manuscript following your input. In the attached pdf file we present reviewer's suggestions and relative answers. We also attached in figure 1 the flowchart based on the suggestion "The authors can add a better flowchart that shows the missing data can be replaced by the other, taking into account the image data sorted from high resolution to low resolution"

Please also note the supplement to this comment:
https://www.nat-hazards-earth-syst-sci-discuss.net/nhess-2017-420/nhess-2017-420-AC1-supplement.pdf

[Figure]

Free satellite data

YES                                                          NO

Co-flood image          Post-flood image          On-demand image

Cloud or        Day and cloud        Day and cloud
Night              free                      free
SAR data          Multispectral data                     Very High resolution data

high-resolution     high-resolution      low-resolution           low cost          Commercial satellite
( Sentinel-1)     (Landsat-8; Sentinel-2)   mapping (MODIS)       aerial platform    (SAR / Multispectral)

**1** Rapid Delimitation of flooded area

On-demand image

Very High resolution photo
low cost  aerial platform

**2A** Flood mapping at very high resolution  only        **2B** Flood mapping at very high resolution on
on delimited areas                                                large area

DTM and ancillary data

**2C** Water depth map

On-demand data and field survey at urban scale

Ultra High resolution mapping using low-cost solution: RPASS photo,
Ground Car Photo and  SfM to asses water level

Legend

Data

Result

**3** Flood mapping, damages assessment  and water depth model at
urban scale

**Fig. 1.**

**Supplement:**

The manuscript includes an original study on flood mapping using various remote sensing image sources and techniques. Therefore it has practical significance. In the literature, as also referenced in the study, there are so many research articles studying the evaluated data types and the techniques, however this study uses most of the available data sources and techniques for a single case showing the efficiency of results. Therefore, a comparative study in which the results of maps using optical and SAR images processed with different remote sensing techniques is presented.

In general, the proposed approach was explained well, the experiments were conducted properly, and the results were discussed in the manuscript. However, there still exists some missing points in the manuscript in terms of the completeness of the paper. Therefore, if they are corrected considering the minor issues highlighted below, the article is recommendable for publication.

Reviewer recommends: Minor revision

**R: we would like to thank you the Reviewer for the detailed revision and his important suggestions. We improved the manuscript following your input. In the following, we present reviewer's suggestions and relative answers.**

**In section 3.1.1**

1) It is noted that available COSMO-SkyMed image has been classified into three main land cover classes as; water-covered areas, i.e., flooded (low amplitude), urban areas (high amplitude) and soil/vegetation (intermediate amplitude). There, what is the type of the classification method used? The result of the accuracy assessment of the classification process was not given?

**R: thank you for the request. We have improved the manuscript with a more detailed explanation. In particular, we added:** *"The Cosmo-Skymed image provided is a simple, not-geocoded image in grayscale format (0-255). After the geocoding we re-classify the SAR amplitude images using empirical thresholds in three main classes: water covered areas (0-60) soil/vegetation (60-160) and urban area (160-255). The investigated area is almost flat, so it is not affected by problems related to geometrical distortions. The validation of the data accuracy was made by comparing the reclassified image with aerial photos, optical images, and land-use."*

2) Authors are recommended to give at least the overall accuracy of the classification! Please also note that in section 4.1.1, the classification accuracy of COSMO-SkyMed was not presented.

**R: We verified the accuracy in terms of classification reliability of this method using aerial photo and CORINE land-use. We add also the table 6 where is resumed accuracy in terms of flooded area detection. About the section 4.1.1 We modified the text (line 346):**

*I) Co-flood mapping, reclassified amplitude of COSMO-SkyMed data. Results of image classification are shown in Figure 3A, where three classes of SAR amplitude were defined by means of empirical thresholds: i) low that correspond to water covered area (blue); ii) intermediate like soil/vegetation (green); iii) high that are urban areas (pink). In the figure are also overlapped the quarry lake from ancillary data (cyan). The accuracy in the correct detection of land-use type is quite good ranging from 80 % for soil and vegetation, 67% for urban area to 61% for water body (tested in quarry lakes). Vegetation and buildings are factors that reduce the detection of water covered areas even using a full-resolution images and more complex processing (Pierdicca et al., 2018). In a second step we selected with a GIS query the low resolution (water covered) class that mostly correspond to the inundated areas and we compared with the it real flooded area. Also the accuracy in the correct detection of flooded areas is quite good: it ranges from 57 % in the lower Oitana stream to 2% in the Po area near Moncalieri. This is related to the time of satellite acquisitions (05:05 UTC of 26 November 2016) some hour before the flood peak. This can be appreciated especially along the Po river, where upstream (near Pancalieri) about the 40% of flooded area was detected, while downstream (Carignano) decrease to 10%. The urban area of Moncalieri limits the capability detection of inundated areas. The false positive errors are less than 5% of the area.*

**In Section 3.1.2**

3) Did the authors apply atmospheric correction to MOSDIS data?
**R: this is an important question. In the first version of the manuscript we didn't apply atmospheric corrections, but then we searched for already corrected product and we made a comparison with the original dataset. In particular, we used MYD09 processed images:** *(Vermote E. - NASA GSFC and MODAPS SIPS - NASA. (2015). MYD09 MODIS/Aqua L2 Surface Reflectance, 5-Min Swath 250m, 500m, and 1km. NASA LP DAAC. http://doi.org/10.5067/MODIS/MYD09.006).* **We compared the corrected images with the previous ones and, since the study area is small and the available atmospheric parameters for correction have 1 km of spatial resolution (water vapour, ozone or aerosol), we did not find significant changes.**
**We wrote also in the manuscript that no further atmospheric correction was applied to MODIS image.**

4) It was also noted that a supervised classification was applied to MODIS by SAGA-GIS. Which supervised classification method was used? Quantify the accuracy of the classification result.
**R: We used Maximum Likelihood method with absolute probability reference. For this revisions we refined the classification, using corrected images, and we also add spectral angle classification. To answer to the question related to accuracy, in terms of flood detection capacity, please see new table 6. In the manuscript line 218 we re-write as follow:**
*Supervised classification of co-flood image. Supervised classifications has already been used in literature to map flooded areas, using machine learning, as described in Ireland et al., (2015). In our work we made a simple supervised classification with SAGA GIS. We first manually defined the training areas with principal land use typologies visible on the false colour image. We try different methodologies for the classifications and we chose as most accurate the maximum likelihood with absolute probability reference and spectral angle methods. We validate the reliability of these classifications with a comparison with false colour image and land-use database. Then we using a GIS query extracted the category "area covered by water or wetland" that mostly correspond to the flooded area for accuracy statics reported in result chapter.*

**In Section 3.2.1 and 3.2.2**

5) DSM is generated from high resolution images. Digital Surface Model (DSM) is not a Digital Terrain Model (DTM). Authors should know the difference between surface and terrain model.
**R: Yes. We used LIDAR DTM downloaded from Regione Piemonte for our Water Depth models (section 3.3). The Digital Surface Model (DSM) were produced by us from SfM processing of aerial and UAV images and we used DSM for 3D model and for the detection of geomorphological features but as a base layer for WD model. We checked in the manuscript if the terms DTM and DSM were properly used.**

**In section 3.3**

6) How did the authors perform water level measurements by GPS-RTK positioning? Give a little detail.
**R: We modified the text to fix this issue: we validated and integrated SfM measures of water level using manual measurement of water level geolocated with high precision using a GPS-RTK positioning.**

**In section 4.1.2**

7) In Figure 4, in the figure caption, the letter of the final item D) appears as C) second time! Correct it:
**R: Ok we corrected it and a new version of figure 4 was made (see below)**

[Figure]

Figure 4: MODIS Aqua satellite image A) False colour band composition 7-2-1 acquired the 12 November 2016.; B) False colour band composition 7-2-1 acquired the 26 November 2016.; C) automatic detection of the flooded area using $MNDWI_{var}>0.3$; automatic detection of the flooded area using supervised classification with maximum likelihood method (D) and with spectral angle method (E). The red box identifies the local case history of Moncaleri (1) and Pancalieri (2)

8) It is observed that MODIS image is classified in to Cloud, Water, Wet soil, Vegetation, and Bare soil whereas COSMO-SkyMed image has been classified into three main land cover classes as water-covered areas (i.e., flooded), urban areas, and soil/vegetation. It looks like only a GIS query can be done between the classes "Water" and "Water-covered areas" classes derived from CSKM and MODIS images, respectively. Authors need to explain in detail how they used maps generated from the classified images.

**R: Yes, we used a GIS query to selected the flooded pixels, but for each dataset the query is based on different criteria; in particular:**

1. **In the case of supervised classifications of MODIS data, flooded pixels correspond to water or wet soil classes. We modify the manuscript (line 375) as follows:**

   *"Supervised Classification. We also made a supervised classification of 26 November MODIS image using maximum likelihood (MLC) (Fig. 4 D) and spectral angle (SA) (Fig. 4 E) methods. In the study area, we classified, four primary land cover: vegetation, bare soil, cloud, and water body / wet soil that almost identify the flooded sector (the water bodies likes the quarry lakes are too small for MODIS pixel). After a visual checking of the classification reliability, we used a GIS query to select the "water covered and wet areas" classes. The query creates a boolean rasters of flooded areas. The accuracy of flood map based on supervised classification is good: it identifies most of the flooded areas for Po river (> 70 %) with low false positive pixel (table 6). Worst results for the are flooded by Chiosla and Oitana.*

2. **In the case of bands ratio (NDVI, MNDWI) made with Sentinel-2 and MODIS DATA, we adopted numerical thresholds empirically based. In the manuscript we write line 369** "*In figure 4 C we identified flooded area using a GIS query with the value MNDWIvar≥0.3. This value is an empirical threshold that select most of detectable flooded area and minimizes false positive errors.*" **And line 383:** "*For both indexes we used GIS queries with empirical thresholds to extract the flooded area:*"

3. **In the case of CSKM, we better specified in the text and in figure 3 that SAR Amplitude Image of CSKM was divided in three classes, based on empirical numerical thresholds, that correspond to different land-use: low (water covered area), medium (soil and vegetation) and high (urban**

**areas). We assumed that water covered areas are almost flooded areas. We modify the manuscript (line 346) as already reported in reply to the comment (2).**

**In addition, in the introduction of par 4.1 (line 339) we better write how we have generated maps from classified images:**

"*The flooded area limits were manually extrapolated considering satellite data and geomorphological features obtained using the hillshade model derived from 5-m DTM of Regione Piemonte and used as a benchmark for the evaluation of the performance of remote sensing analyses. For Po and part of Chisola, the flooded areas were mapped also with help of water height simulation on the base of DTM.*"

**Is now changed as follow:**

"*We manually extrapolated the flooded area perimeters considering both satellite data and geomorphological features observed in the hillshade model derived from 5-m DTM of Regione Piemonte. For the evaluation of automatic flooded area maps based on satellite data we applied a GIS query for each map to create boolean rasters of flooded / not flooded area. Then we overlap the raster with manual polygon for a geo-statistical analysis and accuracy evaluation as reported in table 6.*"

**In Section 4.2.3**

9) Authors declared that " *During the post processing, we realized that the quality of the images extracted from the video was insufficient for the SfM application. For this reason, after a month we performed a second survey along the same path*" Explain the insufficient qualifications for the extracted images used for the SfM application.

**R: Yes, we add the following sentence: "**: …**the bitrate was too low and the frames are too pixelated**. For this reason, after a month we performed a second survey **with a higher bit rate** along the same path, but only six marks still visible (Fig 10 A)".

**In Section 5**

10) It was written that ..........." *the combined used of InSAR data of Sentinel-1 and Cosmo, and multispectral data of MODIS-Aqua and Sentinel-2 allowed creating maps of the flooded area. InSAR data showed a good performance in the real time flood mapping while are weaker for post event mapping.........*" Here, instead of InSAR data the use of SAR data is recommended. It is because, the only amplitude value of the SAR data was used and no interferometric process was applied.

**R: Yes it is true, we have corrected it.**

**In the discussion and results section**

11) Rather than using expressions such as "good agreement", "more precision", "good accuracy", etc; quantify the accuracy or the quality of maps, results, etc .

**R: Thank you for your suggestion, we add some quantitative evaluation of quality of the maps in this section. At end of section 4.1 we also add the table 6 that quantifies the accuracy in flood detection for the automatic processing that we used**

**Table 6. Accuracy in automatic flooded and not flooded area detection**

| Sector | Area km² | Sentinel-2 | | MODIS-Aqua | | | CSKM | Sentinel-1 |
|---|---|---|---|---|---|---|---|---|
| | | MNDWI$_{var}$ | NDVI$_{var}$ | MNDWI$_{var}$ | MLC | SA | Recl Ampl | Δσ° |
| Not Flooded | 259.5 | 87% | 87% | 91% | 94% | 95% | 96% | 99% |
| Flooded area | | | | | | | | |
| - Po | 47.8 | 48% | 37% | 49% | 70% | 64% | 23% | 4% |
| - Oitana | 11.6 | 49% | 42% | 60% | 11% | 36% | 37% | 1% |
| - Chisola | 7.3 | 21% | 51% | 30% | 24% | 23% | 12% | 1% |
| - Chisola urban | 1.1 | 4% | 24% | | | | | |

12) Last but not least, the difficulty of this study is that the satellite data might have not always been received at the time of the hazard occurred! The authors can add a better flow chart that shows the missing data can be replaced by the other, taking into account the image data sorted from high resolution to low resolution:

**R: Yes, it is true: the time of satellite pass over the flooded area is a limit especially with fixed revisit time sensors that we decided to use. Following your suggestion, we create a better flowchart in which we purpose the parameters for the choice the data used for flood mapping.**

**In the manuscript we add this chart as figure 13 and we add the section 4.3 in which our model is explained:**

*4.3 A flood mapping strategy flow chart*

*The flowchart in figure 13 shows the approach that we purpose for the choice of instruments and methods to map the flooded areas, based on the results of this study.*

*If free satellite data are available, it is possible to sort them taking into account the parameters of time elapsed from flood and the spatial resolution:*

*I) The priority is to search for co-flood images that allow an easy mapping. In case of night and cloudy conditions it is necessary to use SAR image (Sentinel-1) while for multispectral data acquired during the day the choice is related to spatial resolution: for instance, Sentinel-2 or Landsat-8 data are more resolute than MODIS data.*

*II) In the case we have post-flood satellite pass only multispectral data can be used. Also this case the Spatial Resolution and time elapsed from the flood are the parameters that should drive the choice. The use of post-flood data implies more complicated post-processing (e.g., bands index variation) and with the support of ancillary data to extract the flooded area map. In general, the rapid access to data portal of free satellite data allows to download the data and to make an evaluation of the best solution for the case under study, that not necessarily is the data with high spatial resolution.*

*After this step it is possible to make a first delimitation of flooded areas, that in case good data may be an already corrected and ready to use map. Then it is possible to focus the acquisition of on-demand of high-resolution sensors only in the most critical or unclear areas (case 2A). If we use only on-demand data, without rapid satellite mapping, we could map at high-resolution large areas (case 2B). This solution however implies higher cost. In case of direct mapping at very-high resolution it is better to use low-cost aerial platforms that are more flexible respect to commercial satellites. After the integration with DEM data the water depth model at basin scale (2C) should be the final result of this chain.*

*Urban area flood mapping (3) can be considered a hotspot priority inside the general flood map. It needs a more accurate and high-resolution mapping with use of ground-based measures (like SfM model based on car photo), RPASS survey, and the creation of a water depth model that is essential for a precise flood magnitude assessment.*

[Figure]

**Flood mapping strategy proposed**

---

## Short Comment (SC1) · 7 Feb 2018

The authors address an interesting and important topic in the field of flood emergency. Several studies are developing methods to integrate remotely-sensed data to produce inundation maps and to estimate hydrological parameters at different time and spatial scale. This study focus on the use of low cost datasets for this kind of activities, applying different datasets at different scale to derive maps and useful hydraulic parameters as Water Depth and Water Level. My overall opinion about the paper is good and I think is suitable for publication. However I suggest the authors to point out and better explain some aspects of the analyses.

1. The use of cosmo-sky images at full resolution is nowadays also a low-cost option

and would provide a definitely more accurate mapping of the inundated areas. Why this option has not been considered instead of the 60 m x 60 m images?

2. Please provide more info about the DTM of Regione Piemonte used to calculate WD (for example time of acquisition, errors on z values etc). Furthermore a discussion of uncertainties in WD and WL estimation is needed.

3. In the discussion the authors mention InSAR but they do not perform any InSAR processing. They only mention $\Delta\sigma$o post-pre-flooding as described in the method section. Please explain.

4. At line 509 authors say: "InSAR data showed a good performance in the real time flood mapping while are weaker for post event mapping". It not clear what is intended here for "good performance" and how the performance was evaluated. This aspect needs to be discussed in more detail. 5. In general I think that in the paper some kind of assessment (better if quantitative) of the results is lacking.

---

## Referee Comment (RC3) · S. Manfreda (Referee) · 10 Feb 2018

The authors carried out a comprehensive study on the use and integration of data from multiple sensors for flood mapping. They used some approaches already tested in literature and others more innovative and experimental. In particular, they designed an approach using free or low cost data/sensors that was tested on a real case study considering both urbanized and not urbanized areas.

The work is certainly of interest for the readers of NHESS. Nevertheless, I have a number of comments that may help the authors in improving the final quality of the manuscript.

1) In order to provide enough information to replicate the experiment, I would suggest

to include more details about the methods used for the classification of the satellite images (COSMO-Skymed, Aqua satellite co-flood image).

2) It is notable a relevant amount of manual operations, as stated in several sentences: - Section 3.1 Flood mapping at regional scale with satellite data "For every considered dataset, we produced a map of the flooded area. . . We use a visual-operator approach to map flooded areas as resulted more precise than automatic classifications especially in the case of post-flood images";

- 3.1.2 Multispectral satellite data, I) Medium-Low resolution satellite data: "For the identification of flooded areas, we make the following elaborations: a) False colour image made with combinations of 7-2-1 bands for a visual interpretation of flooded areas";

- 3.1.2 Multispectral satellite data, I) Medium-Low resolution satellite data: "Supervised maximum likelihood classification of co-flood image made with SAGA GIS. We manually defined the training areas with main land use typology visible on the image. . ."

- 3.1.2 Multispectral satellite data, II) Medium-high resolution satellite data: "To detect flooded area, we first made a visual interpretation using images (Sentinel-2 images) with different bands composition of post-flood data."

- 3.2.3 Ground-based ultra-high resolution images: "For the identification and mapping of water levels, the video is analysed and a frame sequence is extracted from it when the operator sees some marks lefts by water over facades"

- 4 Results, Flood mapping from low to medium-high resolutions with satellite data: "The flooded area limits were manually extrapolated considering satellite data and geomorphological features obtained using the hillshade model derived from 5-m DTM. . ."

- 4.1.2 Flood mapping with multispectral data, I) Multispectral low resolution, MODIS-Aqua: "MNDWI variation (MNDWIVAR) at 20 m of spatial resolution. . .. However, like for NDVI, the presence of many areas with positive variations outside the flooded sector

makes more accurate a manual interpretation"

- 4.1.2 Flood mapping with multispectral data, II) Multispectral medium-high resolution post-flood mapping Sentinel-2: "The images of Sentinel-2 were analysed by visual interpretation of RGB composite image and using two different indexes (NDVI - MNDVI) to identify flooded areas shown in figure 5.

Therefore, I am wondering if such an approach might still be considered low-cost and fast, considering the amount of work that needs to be performed by human operators. Also, the reliability and accuracy of the results would significantly depend on the ability and experience of the operator.

3) It is not possible to infer the performances of the methods/data investigated. Please, describe and provide results of any statistical analyses that you performed.

4) Probably, after 8 pages of Materials and Methods and 7 pages of Results, the article would benefit from an expanded discussion, where those data are interpreted. I would try to address the following questions: What is the overall advice (if exists) authors can give to readers for an efficient approach for flood inundation mapping? Since appears that some analyses provided results not accurate or too uncertain or under/overestimation too significant, is any of the tested methods and data less relevant than others? Can any of these methods/data be completely replaced by the information provided (with a higher accuracy) by other analysed methods/data?

Minor comments The paper contains a number of typing errors that requires a careful review of the English. Below some examples that I found while reading the manuscript.

Check the way citations are written in the manuscript. Sometimes "et al" is followed by no full stop and just the comma (Luino et al, 2009) sometimes a semicolon (Wang et al; 2012), sometimes nothing (Boni et al 2016). Other examples in lines 37-38 "Boni et al 2016; Mason et al 2014; Guy et al 2015; Refice et al 2014; Pulvirenti et al; 2011; Clement et al, 2017; Brivio et al; 2002".

Line 42: correct "authorirhyes"

Line 44: it should be "details" (plural) instead of "detail"

Lines 47-48: check subject-verb agreement "A partial solution could be the use of a Remotely Piloted Aerial System (RPAS), that are usually able". A RPAS system is singular. Line 81: check subject-verb agreement "The basin of Po and Tanaro rivers were"

Line 90: check subject-verb agreement "the actual plain (Fig. 1 B and Fig 1 C) correspond to"

Line 92: check english "The plain is marked by the terraces that delimit of actual Po valley..."

Lines 122: "pre-flood', 'co-flood' and 'pre/post-flood' data". I suggest you to remove pre-flood and just leave "co-flood' and 'pre/post-flood' data", since in the following lines you distinguish and explain these two categories.

Lines 125-127: "Using a multi-scale approach, we developed a methodology that considers the progressive use satellites and then high and ultra-high resolution systems for the acquisition of a dataset that can be used to support the identification of water level reached by the flood and occurred damages". I think an "of" is missing before "satellites". I also suggest authors to think about rephrasing or splitting this sentence in two.

Line 130: "quikly indication". Proper spelling is quickly. By the way, I think the adjective form "quick" is the appropriate one. Line 134 and 137: "Orthophoto" instead of "ortophoto".

Line 264: "the system can flight on demand during the flood of immediately after". "Or" instead of "of"

Line 332: "to assess" instead of "to assesses"

Line 363: "The MODIS-Aqua satellite takes an image…during the late morning of November 26, 2016. " "Took", instead of "takes".

Line 426: "mapped" instead of "mapp7ed"

---

## Referee Comment (RC4) · F. Balik Sanli (Referee) · 14 Feb 2018

The issues outlined in my previous review were replied precisely. Authors made meaningful comments to my questions and concerns. All the answers are now presented in the appropriate places of the manuscript. Figures and flowchart were reorganised and are now more clear. The manuscript is now qualified enough to be published in the Journal of Natural Hazards and Erath System Sciences.

---

## Author Comment (AC4) · 19 Mar 2018

Dear Fusun Balik Sanli,

The authors would thank you for your careful revision that helped us to improve the manuscript

---

## Author Response (AR1)

*Reply to Editor*

Thank you again for your interesting submission "Low cost, multiscale and multi-sensor application for flooded areas mapping" as well as for the response to the comments of the referees.
The referees agree that the manuscript is of interest for the readers of NHESS and the flood risk community.
Combining the reports, it appears to be necessary to provide more details about data and data processing steps as well as quantitative information about the performance and accuracy of classifications. Further, the need was stressed to provide a flow chart of the data sources, processing steps and results.

**AC: Dear Kai Schröter, We would like to thank you for the reading of the manuscript discussion.**
**Following the suggestion of the referees we provided to improve the manuscript. In particular, we gave more detail about Como-Skymed and MODIS data processing. We also add more quantitative results in the discussion chapter as required by the referees. We also add a flow chart showing the flood mapping strategy used for this work. (Fig. 13). We also improved figure 4 and figure 9 respect to the first submission.**

In addition, I would like to emphasize that you conduct a thorough English proof-reading and check of correct referencing, e.g. Guy et al. ,2015 is actually Schumann et al., 2015. Please also make sure to correctly and consistently reference online sources.

**AC: We revised English, using an advanced grammar proofreading software and with a careful re-reading and we corrected the wrong references.**

Your responses seem to take-up the constructive comments of the referees and will allow to present your approach and results more clear. Therefore, I decided for minor revision and I will review your revised manuscript.

**AC: In this document we provide the response (AC) point-by-point to the referee's comment (RC). The main change to manuscript are reported in** *italics* **font and also the line and number of the page of the corresponding track change manuscripts are reported (Page X – line XX)**

*Reply to Referee #1*

RC - Reviewer comment; **AC – Authors comment**

The manuscript includes an original study on flood mapping using various remote sensing image sources and techniques. Therefore it has practical significance. In the literature, as also referenced in the study, there are so many research articles studying the evaluated data types and the techniques, however this study uses most of the available data sources and techniques for a single case showing the efficiency of results. Therefore, a comparative study in which the results of maps using optical and SAR images processed with different remote sensing techniques is presented.
In general, the proposed approach was explained well, the experiments were conducted properly, and the results were discussed in the manuscript. However, there still exists some missing points in the manuscript in terms of the completeness of the paper. Therefore, if they are corrected considering the minor issues highlighted below, the article is recommendable for publication.

Reviewer recommends: Minor revision

**R: we would like to thank you the Reviewer for the detailed revision and his important suggestions. We improved the manuscript following your input. In the following, we present reviewer's suggestions and relative answers.**

**In section 3.1.1**

RC -1) It is noted that available COSMO-SkyMed image has been classified into three main land cover classes as; water-covered areas, i.e., flooded (low amplitude), urban areas (high amplitude) and soil/vegetation (intermediate amplitude). There, what is the type of the classification method used? The result of the accuracy assessment of the classification process was not given?

**AC – 1) thank you for the request. We have improved the manuscript with a more detailed explanation. In particular, we added (Page 7 – line 179):** *"The Cosmo-Skymed image provided is a simple, not-geocoded image in grayscale format (0-255). After the geocoding we re-classify the SAR amplitude images using empirical thresholds in three main classes: water covered areas (0-60) soil/vegetation (60-160) and urban area (160-255). The investigated area is almost flat, so it is not affected by problems related to geometrical distortions. The validation of the data accuracy was made by comparing the reclassified image with aerial photos, optical images, and land-use."*

RC - 2) Authors are recommended to give at least the overall accuracy of the classification! Please also note that in section 4.1.1, the classification accuracy of COSMO-SkyMed was not presented.

**AC – 2) We verified the accuracy in terms of classification reliability of this method using aerial photo and CORINE land-use. We also add the table 6 where is resumed accuracy in terms of flooded area detection. About the section 4.1.1 We modified the text (Page 14 line 389):**

*I) Co-flood mapping, reclassified amplitude of COSMO-SkyMed data. Results of image classification are shown in Figure 3A, where three classes of SAR amplitude were defined by means of empirical thresholds: i) low that correspond to water covered area (blue); ii) intermediate like soil/vegetation (green); iii) high that are urban areas (pink). In the figure are also overlapped the quarry lake from ancillary data (cyan). The accuracy in the correct detection of land-use type is quite good ranging from 80 % for soil and vegetation, 67% for urban area to 61% for water body (tested in quarry lakes). Vegetation and buildings are factors that reduce the detection of water covered areas even using a full-resolution images and more complex processing (Pierdicca et al., 2018). In a second step we selected with a GIS query the low resolution (water covered) class that mostly correspond to the inundated areas and we compared with the real flooded area. Also the accuracy in the correct detection of flooded areas is quite good: it ranges from 57 % in the lower Oitana stream to 2% in the Po area near Moncalieri. This is related to the time of satellite acquisitions (05:05 UTC of 26 November 2016) some hour before the flood peak. This can be appreciated especially along the Po river, where upstream (near Pancalieri) about the 40% of flooded area was detected, while downstream (Carignano) decrease to 10%. The urban area of Moncalieri limits the capability detection of inundated areas. The false positive errors are less than 5% of the area.*

**In Section 3.1.2**

RC – 3) Did the authors apply atmospheric correction to MOSDIS data?

**AC – 3) this is an important question. In the first version of the manuscript we did not apply atmospheric corrections, but then we searched for already corrected product and we made a comparison with the original dataset. In particular,**

we used MYD09 processed images: *(Vermote E. - NASA GSFC and MODAPS SIPS - NASA. (2015). MYD09 MODIS/Aqua L2 Surface Reflectance, 5-Min Swath 250m, 500m, and 1km. NASA LP DAAC. http://doi.org/10.5067/MODIS/MYD09.006).* **We compared the corrected images with the previous ones and, since the study area is small and the available atmospheric parameters for correction have 1 km of spatial resolution (water vapour, ozone or aerosol), we did not find significant changes.**

**We also wrote in the manuscript that no further atmospheric correction was applied to MODIS image.**

RC - 4) It was also noted that a supervised classification was applied to MODIS by SAGA-GIS. Which supervised classification method was used? Quantify the accuracy of the classification result.

**AC – 4) We used Maximum Likelihood method with absolute probability reference. For this revisions we refined the classification, using corrected images, and we also add spectral angle classification. To answer to the question related to accuracy, in terms of flood detection capacity, please see new table 6. In the manuscript we re-write as follow (page 9 line 237):**

*Supervised classification of co-flood image. Supervised classifications have already been used in literature to map flooded areas, using machine learning, as described in Ireland et al., (2015). In our work we made a simple supervised classification with SAGA GIS. We first manually defined the training areas with principal land use typologies visible on the false colour image. We try different methodologies for the classifications and we chose as most accurate the maximum likelihood with absolute probability reference and spectral angle methods. We validate the reliability of these classifications with a comparison with false colour image and land-use database. Then we used a GIS query extracted the category "area covered by water or wetland" that mostly correspond to the flooded area for accuracy statics reported in result chapter.*

**In Section 3.2.1 and 3.2.2**

RC - 5) DSM is generated from high resolution images. Digital Surface Model (DSM) is not a Digital Terrain Model (DTM). Authors should know the difference between surface and terrain model.

**AC – 5) Yes. We used LIDAR DTM downloaded from Regione Piemonte for our Water Depth models (section 3.3). The Digital Surface Model (DSM) was produced by us from SfM processing of aerial and UAV images and we used DSM for 3D model and for the detection of geomorphological features but as a base layer for WD model. We checked in the manuscript if the terms DTM and DSM were properly used.**

**In section 3.3**

RC – 6 )How did the authors perform water level measurements by GPS-RTK positioning? Give a little detail.

**AC – 6) We modified the text to fix this issue: we validated and integrated SfM measures of water level using manual measurement of water level geolocated with high precision using a GPS-RTK positioning.**

**In section 4.1.2**

RC – 7) In Figure 4, in the figure caption, the letter of the final item D) appears as C) second time! Correct it:

**AC – 7) Ok we corrected it and a new version of figure 4 was made (see below)**

[Figure]

*Figure 4: MODIS Aqua satellite image A) False colour band composition 7-2-1 acquired the 12 November 2016.; B) False colour band composition 7-2-1 acquired the 26 November 2016.; C) automatic detection of the flooded area using MNDWI$_{var}$>0.3; automatic detection of the flooded area using supervised classification with maximum likelihood method (D) and with spectral angle method (E). The red box identifies the local case history of Moncaleri (1) and Pancalieri (2)*

RC - 8) It is observed that MODIS image is classified into Cloud, Water, Wet soil, Vegetation, and Bare soil whereas COSMO-SkyMed image has been classified into three main land cover classes as water-covered areas (i.e., flooded), urban areas, and soil/vegetation. It looks like only a GIS query can be done between the classes "Water" and "Water-covered areas" classes derived from CSKM and MODIS images, respectively. Authors need to explain in detail how they used maps generated from the classified images.

**AC – 8) Yes, we used a GIS query to selected the flooded pixels, but for each dataset the query is based on different criteria; in particular:**

1. **In the case of supervised classifications of MODIS data, flooded pixels correspond to water or wet soil classes. We modify the manuscript (Page 15 line 453) as follows:**

    *"Supervised Classification. We also made a supervised classification of 26 November MODIS image using maximum likelihood (MLC) (Fig. 4 D) and spectral angle (SA) (Fig. 4 E) methods. In the study area, we classified, four primary land cover: vegetation, bare soil, cloud, and water body / wet soil that almost identify the flooded sector (the water bodies likes the quarry lakes are too small for MODIS pixel). After a visual checking of the classification reliability, we used a GIS query to select the "water covered and wet areas" classes. The query creates a boolean rasters of flooded*

*areas. The accuracy of flood map based on supervised classification is good: it identifies most of the flooded areas for Po river (> 70 %) with low false positive pixel (table 6). Worst results for the are flooded by Chiosla and Oitana.*

2. **In the case of bands ratio (NDVI, MNDWI) made with Sentinel-2 and MODIS DATA, we adopted numerical thresholds empirically based. In the manuscript we write (Page 15 line 417)** "*In figure 4 C we identified flooded area using a GIS query with the value MNDWIvar≥0.3. This value is an empirical threshold that selects most of detectable flooded area and minimizes false positive errors.*" **(Page 16 line 438)** "*For both indexes we used GIS queries with empirical thresholds to extract the flooded area:*"

3. **In the case of CSKM, we better specified in the text and in figure 3 that SAR Amplitude Image of CSKM was divided into three classes, based on empirical numerical thresholds, that correspond to different land-use: low (water covered area), medium (soil and vegetation) and high (urban areas). We assumed that water covered areas are almost flooded areas. We modify the manuscript (Page 14 line 389) as already reported in reply to the comment (2).**

**In addition, in the introduction of par 4.1 (Page 13 line 369) we better write how we have generated maps from classified images:**

"*The flooded area limits were manually extrapolated considering satellite data and geomorphological features obtained using the hillshade model derived from 5-m DTM of Regione Piemonte and used as a benchmark for the evaluation of the performance of remote sensing analyses. For Po and part of Chisola, the flooded areas were also mapped with the help of water height simulation on the base of DTM.*"

**Is now changed as follow:**

"*We manually extrapolated the flooded area perimeters considering both satellite data and geomorphological features observed in the hillshade model derived from 5-m DTM of Regione Piemonte. For the evaluation of automatic flooded area maps based on satellite data we applied a GIS query for each map to create boolean rasters of flooded / not flooded area. Then we overlap the raster with manual polygon for a geo-statistical analysis and accuracy evaluation as reported in table 6.*"

**In Section 4.2.3**

RC - 9) Authors declared that " *During the post-processing, we realized that the quality of the images extracted from the video was insufficient for the SfM application. For this reason, after a month we performed a second survey along the same path*" Explain the insufficient qualifications for the extracted images used for the SfM application.

**AC – 9) Yes, we add the following sentence: (Page 19 line 532)** "*: …the bitrate was too low and the frames are too pixelated. For this reason, after a month we performed a second survey with a higher bit rate along the same path, but only six marks still visible (Fig 10 A)*".

**In Section 5**

RC - 10) It was written that ..........." *the combined used of InSAR data of Sentinel-1 and Cosmo, and multispectral data of MODIS-Aqua and Sentinel-2 allowed creating maps of the flooded area. InSAR data showed a good performance in the real-*

*time flood mapping while are weaker for post-event mapping........."* Here, instead of InSAR data the use of SAR data is recommended. It is because, the only amplitude value of the SAR data was used and no interferometric process was applied.
**AC – 10) Yes it is true, we have corrected it.**

**In the discussion and results section**

RC - 11) Rather than using expressions such as "good agreement", "more precision", "good accuracy", etc; quantify the accuracy or the quality of maps, results, etc .
**AC – 11) Thank you for your suggestion, we add some quantitative evaluation of quality of the maps in this section. At end of section 4.1 we also add the table 6 that quantifies the accuracy in flood detection for the automatic processing that we used**

*Table 6. Accuracy in automatic flooded and not flooded area detection*

| Sector | Area km$^2$ | Sentinel-2 | | MODIS-Aqua | | | CSKM | Sentinel-1 |
|---|---|---|---|---|---|---|---|---|
| | | MNDWI$_{var}$ | NDVI$_{var}$ | MNDWI$_{var}$ | MLC | SA | Recl Ampl | Δσ$^o$ |
| Not Flooded | 259.5 | 87% | 87% | 91% | 94% | 95% | 96% | 99% |
| Flooded area | | | | | | | | |
| - Po | 47.8 | 48% | 37% | 49% | 70% | 64% | 23% | 4% |
| - Oitana | 11.6 | 49% | 42% | 60% | 11% | 36% | 37% | 1% |
| - Chisola | 7.3 | 21% | 51% | 30% | 24% | 23% | 12% | 1% |
| - Chisola urban | 1.1 | 4% | 24% | | | | | |

RC - 12) Last but not least, the difficulty of this study is that the satellite data might have not always been received at the time of the hazard occurred! The authors can add a better flow chart that shows the missing data can be replaced by the other, taking into account the image data sorted from high resolution to low resolution:
**AC – 12) Yes, it is true: the time of satellite pass over the flooded area is a limit especially with fixed revisit time sensors that we decided to use. Following your suggestion, we create a better flowchart in which we purpose the parameters for the choice the data used for flood mapping.**
**In the manuscript we add this chart as figure 13 and we add the section 4.3 in which our model is explained (Page 20 line 567):**

*4.3 A flood mapping strategy flow chart*
*The flowchart in figure 13 shows the approach that we purpose for the choice of instruments and methods to map the flooded areas, based on the results of this study.*
*If free satellite data are available, it is possible to sort them taking into account the parameters of time elapsed from flood and the spatial resolution:*
*I) The priority is to search for co-flood images that allow an easy mapping. In case of night and cloudy conditions it is necessary to use SAR image (Sentinel-1) while for multispectral data acquired during the day the choice is related to spatial resolution: for instance, Sentinel-2 or Landsat-8 data are more resolute than MODIS data.*
*II) In the case we have post-flood satellite pass only multispectral data can be used. Also this case the Spatial Resolution and time elapsed from the flood are the parameters that should drive the choice. The use of post-flood data implies more*

*complicated post-processing (e.g., bands index variation) and with the support of ancillary data to extract the flooded area map. In general, the rapid access to data portal of free satellite data allows to download the data and to make an evaluation of the best solution for the case under study, that not necessarily is the data with high spatial resolution.*

*After this step it is possible to make a first delimitation of flooded areas, that in case good data may be an already corrected and ready to use map. Then it is possible to focus the acquisition of on-demand of high-resolution sensors only in the most critical or unclear areas (case 2A). If we use only on-demand data, without rapid satellite mapping, we could map at high-resolution large areas (case 2B). This solution however implies higher cost. In case of direct mapping at very-high resolution it is better to use low-cost aerial platforms that are more flexible respect to commercial satellites. After the integration with DEM data the water depth model at basin scale (2C) should be the final result of this chain.*

*Urban area flood mapping (3) can be considered a hotspot priority inside the general flood map. It needs a more accurate and high-resolution mapping with use of ground-based measures (like SfM model based on car photo), RPASS survey, and the creation of a water depth model that is essential for a precise flood magnitude assessment.*

[Figure]

Free satellite data

YES — NO

Co-flood image — Post-flood image — On-demand image

Cloud or Night — Day and cloud free — Day and cloud free

SAR data — Multispectral data — Very High resolution data

high-resolution ( Sentinel-1) — high-resolution (Landsat-8; Sentinel-2) — low-resolution mapping (MODIS) — low cost aerial platform — Commercial satellite (SAR / Multispectral)

**1** Rapid Delimitation of flooded area

On-demand image

Very High resolution photo low cost aerial platform

**2A** Flood mapping at very high resolution only on delimited areas — **2B** Flood mapping at very high resolution on large area

DTM and ancillary data

**2C** Water depth map

On-demand data and field survey at urban scale

Ultra High resolution mapping using low-cost solution: RPASS photo, Ground Car Photo and SfM to asses water level

**3** Flood mapping, damages assessment and water depth model at urban scale

Legend
Data
Result

\*\*\*\*\*\*\*\*\*\*

*Reply to Referee #2*

RC - Reviewer comment; **AC – Authors comment**

RC - The authors address an interesting and important topic in the field of flood emergency. Several studies are developing methods to integrate remotely-sensed data to produce inundation maps and to estimate hydrological parameters at different time and spatial scale. This study focus on the use of low-cost datasets for this kind of activities, applying different datasets at different scale to derive maps and useful hydraulic parameters as Water Depth and Water Level. My overall opinion about the paper is good and I think is suitable for publication. However, I suggest the authors to point out and better explain some aspects of the analyses.
**AC - The authors would like to thanks Domenico Campolongo for his useful revision and suggestions. We reply point-by-point in this document**

RC- 1. The use of Cosmo-sky images at full resolution is nowadays also a low-cost option and would provide a definitely more accurate mapping of the inundated areas. Why this option has not been considered instead of the 60 m x 60 m images?
**AC -1. Yes, CSK is a low-cost option, especially on the Italian territory, where the acquisition is also more regular and frequent. Our idea, however, was to use as much as possible FREE-COST satellite data with regular acquisition plans on the whole Earth. When we selected the SAR data to be used, we initially focused on the COSMO-SkyMed sensor because, in our example, the time of the satellite acquisitions were optimal to study the wave of flood. However, the analysis of the backscattered signal as seen by a couple of Sentinel-1 images, acquired before and after (two days later) the flood peak, has allowed us to detect and study the modifications of the terrain backscattered signals in a few isolated areas that were inundated after the flooding peak. This analysis was performed by proper radiometric calibration of the SAR images, ending up with maps of the pre-/post-flooding backscattered signal difference at a spatial resolution of about 20 m x 20 m. Even though a simple CSK preview image was used, the capability to detect the flooded areas was fully preserved. This demonstrates that the detection capability of the inundated areas and the water level is not significantly impaired by using a low-resolution (60 m x 60 m) SAR image. Of course, if we had used a full resolution CSK data, the mapping would have been much more precise in terms of spatial resolution, but with relatively few improvements in terms of detection. See for instance the document at the following link http://emergency.copernicus.eu/mapping/system/files/components/EMSR192_07TORINOSOUTH_DELINEATION_OVERVIEW_v1_100dpi.pdf describing an experiment where a CSKM full resolution image was used to map the flooded area south of Turin.**

RC - 2. Please provide more info about the DTM of Regione Piemonte used to calculate WD (for example time of acquisition, errors on z values etc). Furthermore a discussion of uncertainties in WD and WL estimation is needed.
**AC -2. The DTM-Lidar was acquired in 2009-2010, the metadata (in Italian) can be found here http://www.geoportale.piemonte.it/geonetworkrp/srv/ita/metadata.show?id=2552&currTab=rndt.**

The accuracy of elevation ranges from +/-0.3 m to +/- 0.6 m in urban areas. This accuracy is quite good for our model, and no better DTMs on the whole area are freely available. The uncertainties in our model are more complicated to quantitative evaluate because they depend on many factors. The number of ground-based WD measures as well as their reliability and geolocation represent the main limitations. The interpolation to obtain water table raster is also another source of error. For instance, in the case of Moncalieri where we have good and controlled measurement points, the error can be estimated in the range of +/-0.2 m. On the rest of Po valley the error is greater than 0.5 m. To minimize the errors, we have made several interpolations to detect the best water table raster that defines the real flooded area.
In the manuscript we have indicated the DTM accuracy and spent a few words on the model uncertainty.

RC -3. In the discussion the authors mention InSAR but they do not perform any InSAR processing. They only mention Δσο post-pre-flooding as described in the method section. Please explain.
AC -3. We changed the manuscript to explain our results, better. In the discussion section, we have added **(Page 22 line 504):** *"We compared pre- and post-flood SAR images of Sentinel-1 making SAR backscattering difference of radiometrically calibrated images. For CSK, we reclassified a simple low-resolution image acquired close to co-flood time. The results show that the timely acquisition of satellite data in the case of a flood event is fundamental: in the areas covered by water (like for CSK data) up to 40% of pixels were correctly classified as flooded and it was possible to detect a clear pattern. On the other hand, SAR is weaker for post-event mapping: in our case, the available data acquired two-three days after the flood (Sentinel-1) support the detection of less than 4% of the flooded area."*

RC -4. At line 509 authors say: "InSAR data showed a good performance in the real-time flood mapping while are weaker for post-event mapping." It is not clear what is intended here for "good performance" and how the performance was evaluated. This aspect needs to be discussed in more detail.
**AC -4. As presented in the comment to reviewer 1, in the revised version we have added table 6 where some quantitative evaluation regarding flood detection accuracy/performance have been presented and discussed. We evaluated the performance making a ratio between the flooded detected by automatic processing of SAR data and the real flooded area. For instance, CSK detected 23 % (but higher upstream up to 50% detection) of the area flooded by Po, whereas Sentinel-1 reach only 4%. The false positive cases (not flooded area classified as flooded) were also evaluated in the accuracy assessment (SAR data have less than 5 % false positive).**

RC -5. In general I think that in the paper some kind of assessment (better if quantitative) of the results is lacking
**AC -5 As introduced in the previous comment, we have added table 6 where we reported some quantitative evaluation of satellite data results. In the manuscript, we have also added more details about the validation process of our results. Some other quantitative data about flood extension and water depth model results in the study area have been added to the discussion/conclusion section.**
**In addition, as suggested by the reviewer 1, we have added a flow chart that shows our approach for mapping flooded areas. This flowchart is based on the results of our study, but we hope that the schema can be considered for a more general approach for low-cost flood mapping**

**You can find the new flowchart at the following link:**
**https://www.nat-hazards-earth-syst-sci-discuss.net/nhess-2017-420/nhess-2017-420-AC1-supplement.pdf**

*Reply to Referee #3*

RC - Reviewer comment; **AC – Authors comment**

RC - The authors carried out a comprehensive study on the use and integration of data from multiple sensors for flood mapping. They used some approaches already tested in literature and others more innovative and experimental. In particular, they designed an approach using free or low-cost data/sensors that was tested on a real case study considering both urbanized and not urbanized areas.

The work is certainly of interest for the readers of NHESS. Nevertheless, I have a number of comments that may help the authors in improving the final quality of the manuscript.

**AC - The authors would like to thanks Salvatore Manfreda for his useful revision and suggestions. We reply point-by-point in this document.**

RC - 1) In order to provide enough information to replicate the experiment, I would suggest to include more details about the methods used for the classification of the satellite images (COSMO-Skymed, Aqua satellite co-flood image).

**AC –1. thank you for your suggestion. As also suggested by another reviewer, we improve the description of the type of data and processing used for COSMO-Skymed and MODIS-Aqua satellites. In particular:**

**a) For Cosmo-data in section 3.1.1 (Page 7 line 179):**

**"***The Cosmo-Skymed data provided is a simple, not-geocoded, image in greyscale format (0-255). After the geocoding, we re-classify, using GIS software, the SAR amplitude images using empirical thresholds in three main classes: water covered areas (0-60) soil/vegetation (60-160) and urban area (160-255). The investigated area is almost flat, so it is not affected by problems related to geometrical distortions. The validation of the classification accuracy was made by comparing the reclassified image with aerial photos, optical images, and land-use.*"

**b) For MODIS-Aqua: for this the revisions we used the atmospheric calibrated data and we added a sentence to the manuscript (section 3.1.2) (Page 8 line 224):**

**"***For the elaboration, we used the MYD09 - MODIS/Aqua Atmospherically Corrected Surface Reflectance 5-Min L2 Swath 500m, (Vermote, 2015) downloaded from* [http://ladsweb.nascom.nasa.gov/](http://ladsweb.nascom.nasa.gov/) *)*"

**We also better explain how we made the supervised classification with this new paragraph (Page 9 line 237):** "*Supervised classification has already been used in literature to map flooded areas, using machine learning, as described in Ireland et al., (2015). In our work we made a simple supervised classification with SAGA GIS. We first manually defined the training areas with main land use typologies visible on the false colour image. We tried different methodologies for the classifications and we chose as most accurate the maximum likelihood with absolute probability reference and spectral angle methods. We validate the reliability of these classifications with a comparison with false colour image and land-use database. Then, using a GIS query, we extracted the category "area covered by water or wetland" that mostly correspond to the flooded area for accuracy statics reported in the result section.*"

RC - 2) It is notable a relevant amount of manual operations, as stated in several sentences: - Section 3.1 Flood mapping at regional scale with satellite data "*For every considered dataset, we produced a map of the flooded area: We use a visual-operator approach to map flooded areas as resulted more precise than automatic classifications especially in the case of post-flood images*";

3.1.2 Multispectral satellite data, I) Medium-Low resolution satellite data: "F*or the identification of flooded areas, we make the following elaborations: a) False colour image made with combinations of 7-2-1 bands for a visual interpretation of flooded areas*";
3.1.2 Multispectral satellite data, I) Medium-Low resolution satellite data: "*Supervised maximum likelihood classification of co-flood image made with SAGA GIS. We manually defined the training areas with main land use typology visible on the image..*"
3.1.2 Multispectral satellite data, II) Medium-high resolution satellite data: "*To detect flooded area, we first made a visual interpretation using images (Sentinel-2 images) with different bands composition of post-flood data.*"
3.2.3 Ground-based ultra-high resolution images: "*For the identification and mapping of water levels, the video is analysed and a frame sequence is extracted from it when the operator sees some marks lefts by water over facades.*"

4 Results, Flood mapping from low to medium-high resolutions with satellite data: "*The flooded area limits were manually extrapolated considering satellite data and geomorphological features obtained using the hillshade model derived from 5-m DTM..*"
4.1.2 Flood mapping with multispectral data, I) Multispectral low resolution, MODIS-Aqua: "*MNDWI variation (MNDWIVAR) at 20 m of spatial resolution: However, like for NDVI, the presence of many areas with positive variations outside the flooded sector makes more accurate a manual interpretation.*"
4.1.2 Flood mapping with multispectral data, II) Multispectral medium-high resolution post-flood mapping Sentinel-2: "T*he images of Sentinel-2 were analysed by visual interpretation of RGB composite image and using two different indexes (NDVI - MNDVI) to identify flooded areas shown in figure 5.*

Therefore, I am wondering if such an approach might still be considered low-cost and fast, considering the amount of work that needs to be performed by human operators. Also, the reliability and accuracy of the results would significantly depend on the ability and experience of the operator.

**AC -2. Thank you for your suggestions. In the following our reply:**

**a) About the cost: We considered this approach low-cost because we used only free satellite data for regional mapping. With actual revisit frequency of free sensors in most of the events, it should be possible to avoid or limit the on-demand commercial satellite or traditional aerial flight over a large area that have high costs and not always can be planned.**
**For instance, for Piemonte flood the cost of traditional aerial survey was about 80'000 € (about 130 €/km$^2$) http://www.regione.piemonte.it/governo/bollettino/abbonati/2017/28/attach/dda1800000620_660.pdf (Regione Piemonte, 2017 Italian)**
**Where is necessary to have a high-resolution mapping, we proposed low-cost (respect to traditional methods) sensors. Go-pro cameras or the RPASs now have affordable costs. Also the aerial photos that we used have a low cost compared to the traditional aerial platform.**

b) About the rapidity in flood mapping: We agree that our manual approach cannot be fast as an automatic classification mapping like the EMSR service, but our aim is different from providing an early warning /emergency mapping that is not validated and represents the inundated area at a specific instant.

Our method has the aim to provide (low cost) maps of the flooded area and water depth with good accuracy and with a reliable validation. These maps, like the maps provided by official authority (e.g., ARPA Piemonte in our case) could be used for a post-flood damages assessment or to improve urban planning and to evaluate damages.

Free satellite images are available few hours or at least one day after their acquisition. Moreover, it is possible to know in advance the time of satellite pass. The processing both for SAR and multispectral satellite data could be made in few days like the water depth model based on DEM.

It is possible to estimate that within few weeks after the flood to have a good map of the flooded area.

At local scale, RPAS, aerial photo and Car Camera surveys can be made in few days, while post-processing and SfM elaboration and data validation require few weeks of work.

c) About human operator: the human factor (operator ability) is crucial, but our method is proposed for people who have expertise in flood mapping (e.g., geomorphologists or remote sensing operator who work in regional services, academia,). Moreover, our methods are mostly based on simple raster GIS calculations that can be easily replicated.

The automatic detection of flooded area works only if we have perfect co-flood image, otherwise an interpretation is necessary. This analysis takes into account also local conditions (geomorphology of flooded are, anthropic structure).

RC - 3) It is not possible to infer the performances of the methods/data investigated. Please, describe and provide results of any statistical analyses that you performed:

AC -3. We thank for your suggestion. In the answer to reviewer 1 we presented a new table (Table 6) in which we show the performances of data and methods that we used.

| Sector | Area km$^2$ | Sentinel-2 MNDWI$_{var}$ | Sentinel-2 NDVI$_{var}$ | MODIS-Aqua MNDWI$_{var}$ | MODIS-Aqua MLC | MODIS-Aqua SA | CSKM Recl Ampl | Sentinel-1 $\Delta\sigma^o$ |
|---|---|---|---|---|---|---|---|---|
| Not Flooded | 259.5 | 87% | 87% | 91% | 94% | 95% | 96% | 99% |
| Flooded area | | | | | | | | |
| - Po | 47.8 | 48% | 37% | 49% | 70% | 64% | 23% | 4% |
| - Oitana | 11.6 | 49% | 42% | 60% | 11% | 36% | 37% | 1% |
| - Chisola | 7.3 | 21% | 51% | 30% | 24% | 23% | 12% | 1% |
| - Chisola urban | 1.1 | 4% | 24% | | | | | |

In the manuscript we explain how we evaluated the performance (introduction of chapter 4 paragraph (Page 14 line 371):

*"For the evaluation of automatic flooded area maps based on satellite data, we applied a GIS query for each map to create boolean rasters of flooded / not flooded area. Then we overlapped the obtained raster with manual polygons for a geo-statistical analysis, for each polygon is reported the percentage pixel classified as flooded/not-flooded. The main results are reported in table 6."*

**We also added more quantitative results in section 5 (discussion / Conclusions):**
**About SAR  (Page 22 line 504)**
"*Concerning SAR data, we reclassified a simple preview low-resolution Cosmo-Skymed amplitude image acquired some hours before the co-flood time. The results show that the time of satellite pass is fundamental: if the area is covered by water (like upstream part of Po river) up to 60% of pixels was correctly classified as flooded and it was possible to observe a clear pattern. We compared pre- and post-flood SAR images of Sentinel-1 making SAR backscattering difference of radiometrically calibrated images. The result shows that SAR is weaker for post-event mapping: in our case 3 days after the flood (Sentinel-1) less than 4% of the flooded area is still detectable*"

**About multispectral data (Page 22 line 515)**
"*The low-resolution MODIS image acquired near the co-flood stage allowed a good identification of flooded areas using different methods: MNDWI variation and supervised classifications. The detection accuracy is good especially for the area flood by Po river where about the 70% of the flooded area was correctly identified.*

*Medium-High resolution multi-spectral data have more capability with post-event images. In this work, we tested NDVI and MNDWI variations for the detection of flooded areas based on the comparison of pre- and post- event images. Both methodologies show quite good -performance in cultivated land, (40 % - 45% of accuracy). Here it is possible to detect a clear pattern: inside the inundated area the percentage of pixel classified as flooded is four times greater than in not flooded area. The inundated areas are more difficult to detect in the dense urban area of Moncalieri (only 4% area was correctly mapped).*

*RC - 4)* Probably, after 8 pages of Materials and Methods and 7 pages of Results, the article would benefit from an expanded discussion, where those data are interpreted. I would try to address the following questions: What is the overall advice (if exists) authors can give to readers for an efficient approach for flood inundation mapping? Since appears that some analyses provided results not accurate or too uncertain or under/overestimation too significant, is any of the tested methods and data less relevant than others? Can any of these methods/data be completely replaced by the information provided (with a higher accuracy) by other analysed methods/data?

**AC -4. Thank you for your suggestions. We added a flowchart (figure 13), and a paragraph (4.3) in the manuscript to clarify all these points:**
**1) The most important thing that we would give to readers is that it is not possible to select a priori which type of data/processing is the better for flood mapping. This depends on different factors:**
**1. Time of satellite acquisition respect to the time of flood peak.**
**2. Type of satellite data (SAR / multispectral, spatial resolution)**
**3. Study area features and risk (dimension, cloud cover, land-use and element at risk)**
**4. Affordable cost (e.g., we use commercial satellite data or traditional aerial photo only if they give significant advantages to flood mapping)**
**Another aspect is the data policy. The applied use of free data could encourage the authorities (e.g., The European Union) to make further investment in open data.**

**2) To compare the performance of data and methods would be necessary that all satellites acquired at the same time and this is a rare combination.**
**For instance, in some cases, a 500 m spatial resolution multispectral image acquired at flood peak could be more accurate than a SAR image with 1 m resolution acquired 2 days after the flood. On the other hand on particular area image from a commercial satellite could be the only one that covers the flood peak. In our case Sentinel-1 show low performance the**

**Cosmo not for the data quality (Sentinel-1 at full resolution is far better than a quicklook image) but only for the time of satellite pass.**

**The results of band indexes variation of Sentinel-2 show little better performance of MNDWI respect to NDVI. In urban areas both NDVI and MNDWI performance are very weak (we add this consideration to manuscript)**

**In the manuscript we added a new paragraph 4.3 (Page 20 line 567):**

*"4.3 Flood mapping strategy flowchart*

*The flowchart in figure 13 shows the approach that we purpose for the choice of instruments and methods to map the flooded areas, based on the results of this study. If free satellite data are available, it is possible to sort them taking into account the parameters of time elapsed from flood and the spatial resolution:*

*I) The priority is to search for co-flood images that allow an easy mapping. In case of night and cloudy conditions it is necessary to use SAR image (Sentinel-1) while for multispectral data acquired during the day the choice is related to spatial resolution: for instance, Sentinel-2 or Landsat-8 data are more resolute than MODIS data.*

*II) In the case we have post-flood satellite pass only multispectral data can be used. Also for post-flood data, the spatial resolution and time elapsed from the flood are the parameters that should drive the choice. The use of post-flood data implies more complicated post-processing (e.g., bands index variation) and with the support of ancillary data and DTM to extract the flooded area map. In general, the rapid access to data portal of free satellite data allows to download the data and to make an evaluation of the best solution for the case under study, that not necessarily is the data with high spatial resolution.*

*After this step, it is possible to make a first delimitation of flooded areas, that in case good data may be an already corrected and ready to use map. Then it is possible to focus the acquisition of on-demand of high-resolution sensors only in the most critical or unclear areas (case 2A). If we use only on-demand data, without rapid satellite mapping, we could map large area at high spatial resolution (case 2B). This solution, however, implies a higher cost. In case of direct mapping at very-high resolution, it is better to use low-cost aerial platforms that are more flexible respect to on-demand commercial satellites. The integration with DEM data allows creating the water depth model at basin scale and a further refinement of flooded area maps (2C).*

*Urban area flood mapping (3) can be considered a hotspot priority inside the general flood map. It needs a more accurate and high-resolution mapping with use of ground-based measures (like SfM model based on car photo), RPAS survey, and the creation of a water depth model that is essential for a precise flood magnitude assessment.*

*It is important to remind that is not possible to select a priori which type of data/processing is the better for flood mapping. The best method to use depends on different factors: 1. Satellite acquisition and time elapsed from flood peak; 2. Type of satellite data (SAR / multispectral, spatial resolution); 3. Study area features and risk (dimension, cloud cover, land-use and element at risk); 4. Affordable cost (e.g., we use commercial satellite data or traditional aerial photo only if they give significant advantages to flood mapping)"*

[Figure]

**Figure 13: Flowchart of the proposed flood mapping strategy**

RC - 5. Minor comments The paper contains a number of typing errors that requires a careful review of the English. Below some examples that I found while reading the manuscript.
**AC – 5. Meanwhile the paper was under revision we provided to improve formation and to revise English.**

RC – 6. Check the way citations are written in the manuscript. Sometimes "et al" is followed by no full stop and just the comma (Luino et al, 2009) sometimes a semicolon (Wang et al; 2012), sometimes nothing (Boni et al 2016). Other examples in lines 37-38 "Boni et al 2016; Mason et al 2014; Guy et al 2015; Refice et al 2014; Pulvirenti et al; 2011; Clement et al, 2017; Brivio et al; 2002".

**AC - 6 - We corrected all the reference using the NHESS format "et al.," Also in the reference section, we have checked for a correct alphabetical index and NHESS format.**

RC 8 - Line 42: correct "authorirhyes"
**AC 8 – corrected**

RC 9 - Line 44: it should be "details" (plural) instead of "detail"
**AC 9 - corrected**

RC 10 - Lines 47-48: check subject-verb agreement "A partial solution could be the use of a Remotely Piloted Aerial System (RPAS), that are usually able". A RPAS system is singular.
**AC 10 - corrected in "***Remotely Piloted Aerial Systems***"**

RC 11 - Line 81: check subject-verb agreement "The basin of Po and Tanaro rivers were"
**AC 11 - corrected in basins**

RC 12 - Line 90: check subject-verb agreement "the actual plain (Fig. 1 B and Fig 1 C) correspond to"
**AC 12– we changed in "corresponds"**

RC 13- Line 92: check english "The plain is marked by the terraces that delimit of actual Po valley…"
**AC 13 - we changed in (Page 4  line 95):  "***The fluvial terraces delimit of actual Po valley...***"**

RC 14 - Lines 122: "pre-flood', 'co-flood' and 'pre/post-flood' data". I suggest you to remove pre-flood and just leave "co-flood' and 'pre/post-flood' data", since in the following lines you distinguish and explain these two categories.
**AC 14 - Done**

RC 15 - Lines 125-127: "Using a multi-scale approach, we developed a methodology that considers the progressive use satellites and then high and ultra-high resolution systems for the acquisition of a dataset that can be used to support the identification of water level reached by the flood and occurred damages". I think an "of" is missing before "satellites".
I also suggest authors to think about rephrasing or splitting this sentence in two.
**AC 15 - now is re-write as follows (Page 5 line 130): "***Using a multi-scale approach, we developed a methodology (Fig. 2) that progressively considers the use satellites and then high and ultra-high resolution systems. The aim is the acquisition of a dataset that can be used to support the identification of water depth and extension reached by the flood. The dataset also allowed making a first evaluation of damages both in urbanized and not urbanized areas.***"**

RC 16 - Line 130: "quikly indication". Proper spelling is quickly. By the way, I think the adjec-tive form "quick" is the appropriate one.
**AC 16- we change quick in general**

RC 17 - Line 134 and 137: "Orthophoto" instead of "ortophoto".
**AC 17 - done**

RC 18 - Line 264: "the system can flight on demand during the flood of immediately after". "Or" instead of "of"
**AC 18 - done**

RC 19 - Line 332: "to assess" instead of "to assesses"
**AC 19- done**

RC 20 - Line 363: "The MODIS-Aqua satellite takes an image,, during the late morning of November 26, 2016. " "Took",
instead of "takes".
**AC 20- done**

RC 21 - Line 426: "mapped" instead of "mapp7ed"
**AC 21 – done**

[revised manuscript text omitted]

**Commentato [ND1]:** This part is changed according to the RC -1 of reviewer 1 and RC -1 of reviewer 3

been multi-looked with one look in the azimuth direction and four looks in the range one, and finally geocoded, by converting the maps from radar geometry into Universal Transverse Mercator (UTM) coordinates (zone 32T).

After these pre-processing steps,  to detect land surface changes induced by flooding, we have computed the difference between the post- and the pre--flooding geocoded backscattering coefficient images, and produced the map of the temporal variation of the surface backscattering ($\Delta\sigma^o_{post-pre-flooding}$).

**3.1.2 Multispectral satellite data.**

In this category, we considered both low and medium resolution images. Unfortunately, we found co-flood images only for low-resolution images.

**I) Medium-Low resolution satellite data**. MODIS (Moderate Resolution Imaging Spectroradiometer) is a system of two sun-synchronous, near-polar orbiting satellites called Aqua and Terra that daily acquire images all over the World (Justice et al., 1998). Terra acquires images in the late morning while Aqua in the early afternoon, satellites also have  a night time pass when they acquire in the thermal bands. This repeat frequency does not occur along the same ground track.  , and the repeat cycle along the track is every 16 days.  The high revisit time allows detecting with more probability flood over  vast areas when they still flooded and not covered by cloud. We searched  for first free-cloud MODIS images from Earthdata portal of NASA  (https://worldview.earthdata.nasa.gov; The selected image  was acquired by Aqua satellite  on 26 November 2016 . We used the 6-bands products with a spatial resolution from 250 to 500 m that range from visible to near--infrared (NIR) and shortwave infrared (SWIR) (Table 4). For the elaboration, we used  the MYD09 - MODIS/Aqua Atmospherically Corrected Surface Reflectance 5-Min L2 Swath 500m, (Vermote, 2015) downloaded from http://ladsweb.nascom.nasa.gov/ ). To have a benchmark of the non-flooded situation, we  also used the Aqua satellite image of 12 November 2016, which  was compared with the image taken during the flood. We did not apply an atmospheric correction to images, because the MYD09 product is adequate our aim. Moreover, the study area is small (20 km) and the atmospheric parameters for correction available at 1 km of spatial resolution (water vapour, ozone or aerosol) don't show significant change. For the identification of flooded areas, we make the following elaborations:

a) False colour image made with combinations of 7-2-1 bands for a visual interpretation of flooded areas;

b)  Modified Normalized Difference Water Index variation MNDWI$_{var}$ (Equation 1). The MNDWI allow detecting water masses or soil moisture. In literature, different combinations for this index are presented and discussed (Xu, 2006; Zhang et al., 2016; Gao, 1996). In our study, we used the ratio between B1 (red band) and B7 (Short

**Commentato [ND2]:** This part is changed according to the RC -3 of reviewer 1 and RC -1 of reviewer 3

Wavelength Infrared - SWIR). The difference with a non-flooded situation can be used for identifying changes in soil moisture. We used the results of supervised classification to mask the cloud cover.

$$MNDWIvar = MNDWIpost - MNDWIpre \qquad \text{where } MNDWI = \frac{(B1-B7)}{(B1+B7)} \ (1)$$

c) Supervised  classification of co-flood image. Supervised classification has already been used in literature to map flooded areas, using machine learning, as described in Ireland et al., (2015). In our work we made a simple supervised classification with SAGA GIS. We first manually defined the training areas with main land use typologies visible on the false colour image. We try different methodologies for the classifications and we chose as most accurate the maximum likelihood with absolute probability reference and spectral angle methods. We validate the reliability of these classifications with a comparison with false colour image and land-use database. Then, using a GIS query, we extracted the category "area covered by water or wetland" that mostly correspond to the flooded area for accuracy statics reported in result chapter.

**II) Medium-high resolution satellite data**. Medium-high resolution multispectral satellites (e.g., Sentinel-2 A and B or Landsat 8) have a longer revisit time (from 5 days for the Sentinel-2 constellation to 16 days for Landsat-8) and it is more difficult to have images at the same time of the maximum flood and cloud free. However, by comparing two images acquired before and after a flood event, it is possible to calculate the variation of different indexes related to change in reflectance of the soil or/and of the vegetation. In this way, it is sometimes possible to map the flooded area indirectly (post-flood mapping). In our study area, we used images taken by Sentinel-2 before the flood (2016, November 11) and after (December 1). Sentinel-2 has some bands at 10-m of spatial resolution and some bands and 20-m of spatial resolution resumed in Table 5. To detect flooded area, we first made a visual interpretation using images with different bands composition of post-flood data. The comparison of considered images allowed calculating the difference between two indexes:

1. The Normalized Difference Vegetation Index (NDVI) variation. The NDVI calculated with 10 m of spatial resolution images Sentinel-2 using the near-infrared band (NIR - B8), and the red band (B4). The NDVI is related to the activity of vegetation, and it is possible by calculating its variation (equation 2) to identify the decrease of NDVI values as an effect of inundation on vegetation (Ahamed et al., 2017). The detection of this change allows mapping flooded areas indirectly.

$$NDVIvar = NDVIpost - NDVIpre \qquad \text{where } NDVI = \frac{(B8-B4)}{(B8+B4)} \quad (2)$$

**Commentato [ND3]:** This part was changed on the base of RC - 4 of reviewer 1 and RC -1 of reviewer 3

[revised manuscript text omitted]

ForIn the considered urban area (of Moncalieri),, where satellite data is weakerhave low accuracy and a precise evaluation of
635 water heightdepth is very importantnecessary for flood damages evaluation, the solution is the acquisition ofintegration with ground-based data. In our work, we tested a low-cost solution with a GO-PRO HERO 3+ (Black Edition) camera installed on a car that allowed to make 3D models and to measure the water height reached during the flood. These measures validated with GPS showed a good accuracygood accuracy, but it is necessary to do the survey within few days after the flood when many water signs are visible. A proposal for the future is to use this system during the emergencies, for instance, on civil
640 protection car, to have a map of water depth with a much higher density of points.

A possible idea is to use this system during the emergencies, for instance, on civil protection car, in order to have a map of water height with much higher density of points.

Using these measures and a high-resolution DTM, it was possible to generate a raster model of water depth that has a good match with the ground truth and could be used about +/- 0.2 m of accuracy. We used the results of WD model for the a
645 preliminary evaluation of building damages or . The model could also be used in the future for flood prevention policy. This model can be also used for the estimation of the degree of damage for every building located in the flooded area. or civil protection plans.

Finally, from our work it is also clear the importance to collect ancillary data also from the new sources on the web: the photo and video collected during the flood by simple citizens can be a precious help for the validation of flooded area maps.

650 **Data availability.**

MODIS data were downloaded from WorldviewNASA LAADS - DAAC portal (https://worldview.earthdata.nasa.gov/?)http://ladsweb.nascom.nasa.gov/) link retrieved 21-11-2017 16-02-2018 12/11/2016: MYD09GAMYD09.A2016317.h18v041215.006.2016319100800 view: https://go.nasa.gov/2ymajYo2016319043220.

655  26/11/2016:

[revised manuscript text omitted]
|--------|------|-------------|------------|-------------|-----|-----|-----------|----------|
| Not Flooded | 259.5 | 87% | 87% | 91% | 94% | 95% | 96% | 99% |
| Flooded area | | | | | | | | |
| - Po | 47.8 | 48% | 37% | 49% | 70% | 64% | 23% | 4% |
| - Oitana | 11.6 | 49% | 42% | 60% | 11% | 36% | 37% | 1% |
| - Chisola | 7.3 | 21% | 51% | 30% | 24% | 23% | 12% | 1% |
| - Chisola urban | 1.1 | 4% | 24% | | | | | |

**Commentato [ND10]:** This table added according to the RC -11 of reviewer 1

[revised manuscript text omitted]

---

## Author Response (AR2)

**Reply To Editor NHESS-2017-420**

Dear Davide Notti and co-authors,

thank you very much for submitting the revised version of your manuscript.

I think all reviewer comments and your suggested changes to the manuscript have been implemented in a satisfactory manner.

However, reading through the manuscript it becomes apparent that English writing needs further improvement. This concerns grammar, terminology and spelling. Please consider proof reading by a native speaker.

**R:** The English writing is now revised by the America Journal Expert service. We add the certificate in the attached PDF

Please harmonize writing and use of abbreviations (e.g. SfM, sfm; façades, facades) **R: done, we used: SfM, facades, RPAS**

Please harmonize referencing to Figures (Fig. or Figure or Fig), and check the sequence of referencing to Figures.

R: Done, we used (Fig. )

specific comments:

L 397 Consider if 'Supervised Classification.' can be deleted.

**R: Removed**

L403 The last sentence is incomplete.

R: Changed in "We obtained the worse results for the area flooded by the Chiosla and Oitana streams"

L 465 ad -> as

**R:** Corrected

LL 500 - 502: how can infer an exponential decrease from two instances in time? please consider rephrasing. **R: We rewrite** "*greatly decreases*" instead of "*exponential*"

L515 2 - 3 m. The upper limit of the legend is 2m. **R: The upper limit of WD model is 2-4 m, we better specify it in the legend of figure 11.**

L526 units are missing **R: Added the units (m)**

L534 purpose -> propose **R: Done**

LL 577 - 579 This sentence is not clear, please rephrase.

**R:** We rewrite as follow: "For example, the two Sentinel-1 satellites provided free images every six days all over Europe, while other satellites have quite high costs and the acquisition is often on-demand. Moreover, most of the time, the on-demand acquisitions are activated only when authorities activate an emergency procedure (e.g., the EMSR of the European Union or by civil protection).

L 600 and 604 measures - > measurements **R: Done**

Please check if Fig. 2 is still needed after including Fig. 13.

**R:** We think that both figures are necessary because the flow charts, even similar, show different things: The first (Fig. 2) is the methodology used for this work, the second is a general methodological approach proposed for the readers.

caption Fig 6: photo took -> photos taken **R: Corrected**

caption Fig 7: 'allowed' can this be deleted? ... 'cut a:an erosion' Please check. **R: Corrected**

caption Fig 8: 'map of flooded Chisola flooded area' Please check.# **R: Changed in** *"map of areas flooded by Chisola stream"*

Figure 11: In the legend: River embankment ropture -> rupture; the GPS and SfM symbols cannot be seen on the map

R: Corrected, we also improved the visibility of symbols

AMERICAN JOURNAL EXPERTS

**EDITORIAL CERTIFICATE**

This document certifies that the manuscript listed below was edited for proper English language, grammar, punctuation, spelling, and overall style by one or more of the highly qualified native English speaking editors at American Journal Experts.

**Manuscript title:**

Low cost, multiscale and multi-sensor application for flooded areas mapping

Authors:

Daniele Giordan

Date Issued: April 30, 2018

**Certificate Verification Key: 565C-109C-52FC-74DD-39A3**

This certificate may be verified at www.aje.com/certificate. This document certifies that the manuscript listed above was edited for proper English language, grammar, punctuation, spelling, and overall style by one or more of the highly qualified native English speaking editors at American Journal Experts. Neither the research content nor the authors' intentions were altered in any way during the editing process. Documents receiving this certification should be English-ready for publication; however, the author has the ability to accept or reject our suggestions and changes. To verify the final AJE edited version, please visit our verification page. If you have any questions or concerns about this edited document, please contact American Journal Experts at support@aje.com.

American Journal Experts provides a range of editing, translation and manuscript services for researchers and publishers around the world. Our top-quality PhD editors are all native English speakers from America's top universities. Our editors come from nearly every research field and possess the highest qualifications to edit research manuscripts written by non-native English speakers. For more information about our company, services and partner discounts, please visit www.aje.com.

**Low cost, multiscale and multi-sensor application for flooded areas mapping.**

Daniele Giordan1, Davide Notti1, Alfredo Villa2, Francesco Zucca3, Fabiana Calò4, Antonio Pepe4, Furio Dutto5, Paolo Pari6, Marco Baldo1, Paolo Allasia1

[revised manuscript text omitted]

by the Alps-range. This meteorological configuration causes heavy rainfallsrainfall, especially in autumn when the warm Ligurian seaSea is a source of additional energy and humidity (Buzzi et al., 1998; Pinto et al., 2013). In the last 30 years, four main floods hit this region: September 1993 (Regione Piemonte, 1996, ). November 1994 (Luino, 1999), October 2000

(Cassardo et al., 2013) and November 2016 (ARPA Piemonte, 2016)

In November 2016, a severe flood hit the Piemonte region (NW of Italy). In several areas of the Piemonte, in the period 21– \_25 November 2016, different rain gauges registered an amount of rainfall up to 600 mm that represents the, which represented 50-% of the mean annual precipitation (Fig. 1 B). The basins of the Po and Tanaro rivers were the most affected by the flood that was very similar, for rainfall distribution and river discharge, to the 1994 event, which is considered one of the most destructive that has occurred in lastthe past decades (Luino, 1999). This time, the event caused huge damageslarge damage, activated numbers of numerous landslides and debris flows and caused the inundation of large areas. The civil protection system

90 managed the emergency, and the number of victims was sharply reduced compared to the 1994 event that caused 70 victims. The 2016 flood caused a victim in Chisone valley Valley, not far from Torino.

The presented case study area is located in the Po plain south of Turin city (Fig. 1 C). This area is mainly occupied by intensive agricultural activity and urban areas mainly primarily located in the northern part. At To the south of Turin, many industrial and commercial areas were built in the last decades nearbynear the rivers. From the geomorphological point of view, the actual

- 95 plain (Fig. 1 B and Fig. 1 C) corresponds to the fill of the Plio-Pleistocenic Savigliano basin (S.B.), delimited by the western Alps, Turin C) Hills (T.H.) and Poirino Plateau (P.P). In the western part, it is possible to find alluvial fans of the Chisone, Pellice and Chisola streams (Carraro et al., 1995). The fluvial terraces delimit of the actual Po valley Valley with evident relict geomorphology-like, such as paleo-meandermeanders. The anthropic influence is remarkable with like-quarry lakelakes, revetments and embankmentembankments that constrain the riverbeds (Fig. 1-C). The geomorphology is a crucial factor 100 that constrains the flooded area shape and the water height.
- This area was affected by the flooded flooding of the Po River and other tributaries, in. In particular, the Chisola and Oitana streams causing several damages, caused severe damage. The Po river River between the Carignano and Turin stations reached a maximum discharge of 2000–2200 m3/s in the late evening of 25 November 2016. The mean discharge of this monitoring station in November iswas 70 m3/s. The Chisola stream Stream registered a discharge of 200 m3/s (November average 17 m3/s) 105 near Moncalieri inon the afternoon of 25 November (ARPA Piemonte, 2016).

Inside this area (Fig. 1 C), we focused our attention in particular on two sites werewhere high-resolution data were acquired: The village of Pancalieri, located in-on the left side of the Po riverRiver, just after the confluence with the Pellice riverRiver. In this area, it is evident that the presence of ancient Po riverRiver meanders which are present, and they were reactivated by the flood with damagesdamage to some settlements and the destruction of communications roads.

- 110 The town of Moncalieri (about 60'000 approximately 60,000 inhabitants) is located south of Turin in a human-made environment. This area was flooded by Chisola stream Stream on the late morning of 25 November partly due to the collapse of some sections of the river embankment. The water interestedentered many residential, service and industrial areas with a maximum water height of 1.5—2 m. Another sector of the Moncalieri municipality was also flooded by the Po River in the evening of 25 November, with other damagesdamage to commercial and industrial infrastructures.
- 115 The activation of Copernicus Emergency Management Service (© 2016 European Union) EMSR-192 (http://emergency.copernicus.eu/mapping/list-of-components/EMSR192) allowed to mapping of the flooded areas (delineation maps) using Radarsat-2, COSMO-Skymed, and Pleiades images in the most critical areas of the Piemonte. In some areas-like, such as Moncalieri-also, a map of damages the damage (grading maps) was also produced. However, the available delineations maps represent the automatic extraction of the flooded area at the moment of image acquisition, and

120 generally not at the maximum extension. In the case of Piemonte, they cannot be used for exhaustive modelling and  $\frac{\text{damages} \text{damage}}{\text{damages}}$  evaluations. The preliminary estimate of  $\frac{\text{damages} \text{the damage}}{\text{damage}}$  to buildings made by the municipality of Moncalieri is about was approximately 50 million of  $\in (M \in)$  for industrial buildings and 13 M $\in$  for residential buildings, and  $\frac{\text{others} \text{another}}{\text{others}}$  6 M $\in$  for  $\frac{\text{damages} \text{damage}}{\text{
[revised manuscript text omitted]
 areaareas in this study. |                  |                    |                                    |                   |  |  |
|-----------------------------------------------------------------------------------------------|------------------|--------------------|------------------------------------|-------------------|--|--|
|                                                                                               |                  |                    | Covered area by a                  |                   |  |  |
|                                                                                               |                  | Spatial resolution | single scene                       | Min. Revisit time |  |  |
| Туре                                                                                          | Sensor used      | ( m )       | ( km 2 )         | (Day)             |  |  |
|                                                                                               |                  | 1 – Satellite Data |                                    |                   |  |  |
| SAR –X band                                                                                   | COSMO-SkyMed     | 60                 | > <del>1'0001,000</del>     | 4                 |  |  |
|                                                                                               |                  | 5 (ground range) x |                                    |                   |  |  |
| SAR- C band                                                                                   | Sentinel-1A/B    | 20 (azimuth)       | > <del>10'000</del> 10,000         | 6                 |  |  |
| Multi spectral                                                                                |                  |                    | >                                  |                   |  |  |
| Multispectral                                                                          | MODIS-Aqua       | 500                | <del>100'0000100,0000</del> | Daily             |  |  |
| Multi spectral                                                                                |                  |                    |                                    |                   |  |  |
| Multispectral                                                                          | Sentinel-2       | 10 / 20            | > <del>10'000</del> 10,000         | 10 (5)            |  |  |
|                                                                                               |                  | 2- Aerial data     |                                    |                   |  |  |
| Very High res.                                                                                |                  |                    |                                    |                   |  |  |
| visible band                                                                                  | Tecnam P92-JS    | 0.01               | 100 km 2                | On-demand         |  |  |
| Ultra-High                                                                                    |                  |                    |                                    |                   |  |  |
| resolution visible                                                                            | RPASs CarbonCore |                    |                                    |                   |  |  |
| band                                                                                          | 950 octocopter   | 0.02 / 0.03        | $< 10 \text{ km}^2$                | On-demand         |  |  |
| DTM LIDAR                                                                                     | Airborne         | 5                  | Piemonte region                    | Archive data      |  |  |
|                                                                                               |                  | 3- Ground-Based    |                                    |                   |  |  |
| Photo / video from                                                                            |                  |                    |                                    |                   |  |  |
| car platform                                                                                  | GO-PRO HERO 3+   | 0.02 / 0.03        | Local /urban                       | On demand         |  |  |

**Table 2. Resume of satellite data in relation with the flood stage.**

| Satellite  | Spatial    | Acquisition time       |                        |                        |  |
|------------|------------|------------------------|------------------------|------------------------|--|
|            | Resolution | Pre-flood              | Co-flood               | Post-flood             |  |
| COSMO-     | Medium     |                        | 05:05 UTC - 25/11/2016 |                        |  |
| SkyMed     |            |                        |                        |                        |  |
| Sentinel-1 | Medium     | 05:35 UTC - 22/11/2016 |                        | 05:35 UTC - 28/11/2016 |  |
| A/B        |            |                        |                        |                        |  |
| MODIS      | Medium-    | 12:30 UTC - 12/11/2016 | 12:30 UTC - 26/11/2016 |                        |  |
| Aqua       | Low        |                        |                        |                        |  |
| Sentinel-2 | Medium-    | 15:19 UTC - 11/11/2016 |                        | 14:19 UTC - 01/12/2016 |  |
|            | High       |                        |                        |                        |  |

| Table 5. Characteristics of the Sentiner-1 tataset used in this study | acteristics of the Sentinel-1 dataset used in this study |
|-----------------------------------------------------------------------|----------------------------------------------------------|
|-----------------------------------------------------------------------|----------------------------------------------------------|

| Satellite               | Sentinel-1 A/B            |
|-------------------------|---------------------------|
| Sensor Parameter        | C-band 5.405 GHz          |
| Orbit                   | Descending                |
| Pre-flood acquisitions  | 22/11/2016                |
| Post-flood acquisitions | 28/11/2016                |
| Data format             | Single Look Complex (SLC) |
| Azimuth pixel spacing   | ~13                       |
| [m]                     | 15                        |
| Range pixel spacing [m] | ~2                        |

**Table 4. Characteristics of the Aqua MODIS data used in this work.**

| Band | Bandwidth (nm) | Band type | Spatial resolution
(m) |
|------|----------------|-----------|---------------------------|
| B1   | 620 - 670      | Red       | 500                       |
| B2   | 841 - 876      | NIR       | 500                       |
| B3   | 459 - 479      | Blue      | 500                       |
| B4   | 545 - 565      | Green     | 500                       |
| B5   | 1230 - 1250    | SWIR      | 500                       |
| B7   | 2105 - 2155    | SWIR      | 500                       |
|      |                |           |                           |

Table 5. Characteristics of Sentinel-2 data used in this work.

| Band | Wavelength  | Band type | Spatial resolution |
|------|-------------|-----------|--------------------|
|      | (nm) |           | (m)                |
| B2   | 490         | Blue      | 10                 |
| B3   | 560         | Green     | 10                 |
| B4   | 665         | Red       | 10                 |
| B8   | 842         | NIR       | 10                 |
| B5   | 705         | NIR       | 20                 |
| B6   | 740         | NIR       | 20                 |

| B7  | 783  | NIR  | 20 |
|-----|------|------|----|
| B8a | 865  | NIR  | 20 |
| B11 | 1610 | SWIR | 20 |
| B12 | 2190 | SWIR | 20 |

**Table 6. Accuracy in automatic flooded and not flooded area detection**.**

| Sector          | Area
km 2 | Sentin                      | el-2                | MODIS-Aqua           |     | COSMO-
SkyMed | Sentinel-1 |                       |
|-----------------|-------------------------|-----------------------------|---------------------|----------------------|-----|------------------|------------|-----------------------|
|                 |                         | MNDWI var | NDVI var | MNDWI var | MLC | SA               | Recl Ampl  | $\Delta\sigma^{ m o}$ |
| Not Flooded     | 259.5                   | 87%                         | 87%                 | 91%                  | 94% | 95%              | 96%        | 99%                   |
| Flooded area    |                         |                             |                     |                      |     |                  |            |                       |
| - Po            | 47.8                    | 48%                         | 37%                 | 49%                  | 70% | 64%              | 23%        | 4%                    |
| - Oitana        | 11.6                    | 49%                         | 42%                 | 60%                  | 11% | 36%              | 37%        | 1%                    |
| - Chisola       | 7.3                     | 21%                         | 51%                 | 30%                  | 24% | 23%              | 12%        | 1%                    |
| - Chisola urban | 1.1                     | 4%                          | 24%                 |                      |     |                  |            |                       |

 Table 7. Water height measures
 measures
 measures

 with DTM.
 output
 output

| re point
inates | Water Depth (m)                                                                               |                                                                                                                                                         |                                                                                                                                                                                                                                                                                                                                                                                                            |  |  |  |  |
|--------------------|-----------------------------------------------------------------------------------------------|---------------------------------------------------------------------------------------------------------------------------------------------------------|------------------------------------------------------------------------------------------------------------------------------------------------------------------------------------------------------------------------------------------------------------------------------------------------------------------------------------------------------------------------------------------------------------|--|--|--|--|
| UTM Y              | SfM
(+/-( +
0.05)                                                                | GPS                                                                                                                                                     | DTM
(+/-
(±0.2)                                                                                                                                                                                                                                                                                                                                                                                      |  |  |  |  |
| 4983240            | 1.56                                                                                          | 1.60                                                                                                                                                    | 1.61                                                                                                                                                                                                                                                                                                                                                                                                       |  |  |  |  |
| 4983152            | 1.45                                                                                          | 1.40                                                                                                                                                    | 1.42                                                                                                                                                                                                                                                                                                                                                                                                       |  |  |  |  |
| 4982644            | 0.84                                                                                          | 0.78                                                                                                                                                    | 1.01                                                                                                                                                                                                                                                                                                                                                                                                       |  |  |  |  |
| 4981624            | 0.82                                                                                          | 0.81                                                                                                                                                    | 0.78                                                                                                                                                                                                                                                                                                                                                                                                       |  |  |  |  |
| 4981188            | 1.28                                                                                          | 1.35                                                                                                                                                    | 1.56                                                                                                                                                                                                                                                                                                                                                                                                       |  |  |  |  |
| 4980993            | 1.40                                                                                          | 1.37                                                                                                                                                    | 1.34                                                                                                                                                                                                                                                                                                                                                                                                       |  |  |  |  |
|                    | re point
inates
UTM Y
4983240
4983152
4982644
4981624
4981188
4980993 | re point
inates Wate
SfM
UTM Y (+/-(±
0.05)
4983240 1.56
4983152 1.45
4982644 0.84
4981624 0.82
4981188 1.28
4980993 1.40 | re point
inates         Water Depth           Water Depth         SfM           UTM Y         (+/-(±)         GPS           0.05)         4983240         1.56         1.60           4983152         1.45         1.40           4982644         0.84         0.78           4981624         0.82         0.81           4981188         1.28         1.35           4980993         1.40         1.37 |  |  |  |  |

Figure 1. A) Location of Piemonte region in Italy; B) Rainfall in Piemonte region during the 21-25 November flood event (Basedbased on ARPA Piemonte data) and location of study area, S. B. = Savigliano Basin, P. P. = Poirino Plateau, T.H = Turin Hills; C) Detailed view of the study area with discharge in the stream gauge stations and the location of Pancalieri (1) and Moncalieri (2) local areas case history.

Figure 2. The flowchart illustrating the multiscale flood mapping approaches proposed in this work.

Figure 3. A) Reclassified Quicklook Amplitude SAR Image acquired at 05:05 UTC of 25 November 2016 - COSMO-SkyMed© ASI [2016] - B) Sentinel-1 geocoded backscattering coefficient difference ( $\Delta \sigma^{\circ}$ ); C) Example of change backscattering between the preand the post-flood image is detectable.  $\frac{D^{\circ}; D^{\circ\circ}; D^{\circ\circ}; D^{\circ\circ}}{D^{\circ\circ}; D^{\circ\circ}}$  detail of some areas where  $\Delta \sigma^{\circ}$  still detect water.

Figure 4. MODIS Aqua satellite image A) False colour band composition 7-2-1 acquired during the 12 November 2016; event; B) False colour band composition 7-2-1 acquired during the 26 November 2016; event; C) automaticAutomatic detection of the flooded area using MNDWIvar >  $_{o}$  0.3; automatic detection of the flooded area using supervised classification with maximum likelihood method (D) and with spectral angle method (E). The red box identifies the local case history of Moncaleri (1) and Pancalieri (2).

920 Figure 5. Sentinel-2 analysis and validation: A) NDVI variation 10 m of spatial resolution, B) MNDWI variation 20 m of spatial resolution, C) Simulation of flooded area water depth for the Po riverRiver based on 5-m DTM and river height level registered in the Arpa Piemonte stream gauge.

Figure 6. A) Aerial photo at 10 cm of resolution taken the 28 November 2016; B and C) photosPhotos taken from a local newspaper and geolocalized with Google Streetview; D) Geomorphological elements and model of estimated water depth (m).

Figure 7. The SfM 3-D model obtained from the very high-resolution aerial photo, near the Po River at the south of Pancalieri. A) Location of 3-D models; -B) The approximate measurmanetmeasurement of water depth on sand deposits of a quarry (A);- C) The effect of meander cut: an erosion channel and the destruction along a stretch of road.

Figure 8. A) mapMap of areas flooded by Chisola stream in Moncalieri municipality with location of detailed photo; B) Aerial photo at 0.1 m of resolution showing the breach of river embankment and the 'alluvial fan' created by water flow showed with a 3-dD view in figureFigure 9; C) RPASs photo at 0.02 m of spatial resolution showing a collapsed wall in the Tetti Piatti area and a deposit of damaged goodgoods from the nearby house.

910

915

930

Figure 9. 3-D Model derived from the RPASs aerial photo and SfM elaborations photo overlap. The model shows the river embankment rupture (point B in figureFigure 8A), the geomorphological effects in the neighbour areas. It is also possible to estimate the water depth from the signs on the embankment.

- Figure 10. A) Map of the flooded area by Chisola streamStream in the Moncalieri municipality with measured water height and location of 3-dD photo; B) Installation of the GO-PRO HERO 3+ (Black Edition) camera and GPS antenna over the car (the processing system for STANAG 4609 encoding was installed inside the car); C and D-) Examples of 3-dD models made with Structure from MotionSfM in which it was possible to measure the water height.
- 945 Figure 11. A) Water depth levee map based on SfM data and 5-m DTM model; B) Water level reached by Chisola streamStream in ARPA Piemonte station; C) Geolocated third part photo: 25/11/2016 aerial view of flooded area findfound on the web (https://viveremoncalieri.it/2016/11/30/3760/ ).

Figure 12. A) Water level height map based on SfM data and 5-m DTM model; C) Zoom on the building of figure D; B and D) Photo where it is possible to observe the water depth from the newspaper "La Stampa" geolocated using Google Street view.

Figure 13. -Flowchart of the proposed flood mapping strategy\_